# Metabolic basis for the evolution of a common pathogenic *Pseudomonas aeruginosa* variant

**Dallas L Mould**[1], **Mirjana Stevanovic**[1], **Alix Ashare**[1,2], **Daniel Schultz**[1], **Deborah A Hogan**[1]*

[1]Department of Microbiology and Immunology, Geisel School of Medicine at Dartmouth, Hanover, United States; [2]Department of Medicine, Dartmouth-Hitchock Medical Center, Lebanon, United States

**Abstract** Microbes frequently evolve in reproducible ways. Here, we show that differences in specific metabolic regulation rather than inter-strain interactions explain the frequent presence of *lasR* loss-of-function (LOF) mutations in the bacterial pathogen *Pseudomonas aeruginosa*. While LasR contributes to virulence through its role in quorum sensing, *lasR* mutants have been associated with more severe disease. A model based on the intrinsic growth kinetics for a wild type strain and its LasR⁻ derivative, in combination with an experimental evolution based genetic screen and further genetics analyses, indicated that differences in metabolism were sufficient to explain the rise of these common mutant types. The evolution of LasR⁻ lineages in laboratory and clinical isolates depended on activity of the two-component system CbrAB, which modulates substrate prioritization through the catabolite repression control pathway. LasR⁻ lineages frequently arise in cystic fibrosis lung infections and their detection correlates with disease severity. Our analysis of bronchoalveolar lavage fluid metabolomes identified compounds that negatively correlate with lung function, and we show that these compounds support enhanced growth of LasR⁻ cells in a CbrB-controlled manner. We propose that in vivo metabolomes contribute to pathogen evolution, which may influence the progression of disease and its treatment.

*For correspondence: dhogan@dartmouth.edu

Competing interest: The authors declare that no competing interests exist.

## Editor's evaluation

This study aimed to identify the genetic foundation favoring selection of lasR mutants in laboratory and clinical isolates from persons with CF. They selected these mutants using a predictable and quantitative framework of evolution experiments and then identified their genetic underpinnings by a suppressor screen. The role of cbrAB as a key intermediate is important and ties together several reports of nutrient-dependent advantages of lasR, including those that may explain their adaptation to conditions found in the CF airway.

## Introduction

Quorum sensing (QS) is a mechanism of microbial communication that regulates the expression of a suite of genes in response to diffusible autoinducers in a population (*Schuster and Greenberg, 2007*; *Schuster et al., 2003*). Despite the importance of cell-cell communication for virulence (*Rumbaugh et al., 2009*) and high conservation across divergent phylogenies, key QS regulators in diverse species, such as *Pseudomonas aeruginosa*, *Vibrio cholerae*, and *Staphylococcus aureus*, frequently lose function (*Mould and Hogan, 2021*), due to recent missense and nonsense mutations, indels, or genome

**eLife digest** Bacteria can evolve quickly, a skill that proves useful in ever-changing environments. For example, individuals in many bacterial species can start to work together under certain circumstances; this ability is underpinned by a system called quorum sensing, which allows cells to detect nearby conspecifics. However, species of harmful bacteria often lose their quorum sensing abilities when they infect humans. This is the case for *Pseudomonas aeruginosa,* which normally lives in the soil but can also cause deadly conditions, especially in hospital settings.

Patients often carry *P. aeruginosa* with mutations that disable the quorum-sensing signal receptor LasR, a molecular actor that can switch on many other genes in a cell. People who are infected with *P. aeruginosa* strains carrying a damaged version of the *lasR* gene are typically more ill and less likely to recover. Why this is the case – and in fact, why genes associated with quorum sensing often lose function during infection – is still unclear.

To investigate this question, Mould et al. used laboratory evolution experiments and computer models of *P. aeruginosa* growth to understand how *lasR* mutant cells evolve. Differences in growth rates and ways to use resources (rather than changes in cell-to-cell interactions) best explained why *lasR* mutants become more successful. Further experiments narrowed down the molecular cascade required for the rise of *lasR* mutants, identifying a pathway that regulates how *P. aeruginosa* switches between different nutrient sources.

This work reveals a new connection between quorum sensing genes and nutrient regulation in bacterial cells. Loss of functional LasR changes the way that cells use nutrients, and thus will reshape how they interact with host cells and other bacteria. This insight could lead to better ways to predict the outcomes of bacterial infections and how to best treat them.

rearrangements. These paradoxical findings suggest that there may be connections between QS and other key physiological pathways that have yet to be revealed.

In *P. aeruginosa,* many isolates from humans, plants, and water sources have loss-of-function mutations in the gene encoding the transcription factor LasR (*Groleau et al., 2022*; *O'Connor et al., 2021*), which is central to an interconnected QS network (*Schuster et al., 2003*). LasR⁻ isolates have been repeatedly observed in *P. aeruginosa* lung infections in people with cystic fibrosis (pwCF) (*Smith et al., 2006*), and LasR⁻ isolate detection is associated with more rapid lung function decline and more inflammation than in comparator populations (*Hoffman et al., 2009*; *LaFayette et al., 2015*). In a clinical study of acute corneal infections (*Hammond et al., 2016*), LasR⁻ strains also correlated with more damage and worse outcomes.

Multiple studies contribute to our understanding of the physiologies and social interactions that impact LasR LOF mutant fitness. Several studies provide evidence in support of the model that LasR⁻ strains are 'social cheaters' that reap the benefits of shared goods secreted by neighboring wild-type cells without incurring the metabolic costs (*Sandoz et al., 2007*). In this case, LasR⁻ strains grow better when the wild type is in the majority, and crash when a critical threshold of LasR⁻ cells is surpassed due to insufficient wild-type support (*West et al., 2006*). The extent of *lasR* mutant 'cheating' depends on the cost-benefit difference, and multiple shared goods, including siderophores, must be considered (*Ozkaya et al., 2018*). To combat the rise of cheaters, *P. aeruginosa* produces products such as hydrogen cyanide, rhamnolipids, or pyocyanin that inhibit the growth of quorum sensing mutants through a process known as 'policing' (*Castañeda-Tamez et al., 2018*; *García-Contreras et al., 2020*; *Wang et al., 2015*). There is evidence that the presence of LasR⁻ subpopulations may be beneficial (*García-Contreras and Loarca, 2020*) and lead to emergent properties including metabolite-driven interactions between wild type and *lasR* mutants that provoke the production of QS-controlled factors by the *lasR* mutant to levels greater than in wild-type monocultures (*Mould et al., 2020*). In addition to the interactions between LasR⁺ and LasR⁻ cells that influence the fitness and behavior of LasR⁻ strains described above, there are important intrinsic characteristics of LasR⁻ strains including increased Anr-regulated microoxic fitness (*Clay et al., 2020*), resistance to alkaline pH in aerobic conditions (*Heurlier et al., 2005*), and altered metabolism (*D'Argenio et al., 2007*). The metabolic advantages associated with LasR⁻ strains include growth on individual amino acids (*D'Argenio et al., 2007*). The numerous differences described between LasR⁺ and LasR⁻ strains indicate that an understanding of the factors

that drive the rise and persistence of *lasR* mutants may be complex and are not yet well understood. While it is clear that there are many ways in which *lasR* LOF mutants differ from their LasR +progenitors, a common trait that promotes the rise of LasR⁻ strains in diverse environments, even in rich and minimal laboratory media (*Heurlier et al., 2005*; *Scribner et al., 2022*; *O Brien et al., 2017*; *Luján et al., 2007*; *Qi et al., 2016*; *Robitaille et al., 2020*; *Sandoz et al., 2007*; *Wong et al., 2012*; *Yan et al., 2018*), has not been established.

Here, we use mathematical modeling, experimental evolution-based genetic screens, phenotype profiling, and whole-genome sequencing of evolved communities in different backgrounds to understand the rise of LasR⁻ strains over only a few serial passages. We identified the CbrAB pathway as the strongest contributor to the rise of *lasR* LOF mutants, and our findings were not specific to strain background or medium. LasR⁻ strains are more commonly detected in samples from individuals with more severe CF lung disease (*Smith et al., 2006*). Analysis of bronchoalveolar lavage samples from pwCF and non-CF comparators identified several compounds that were higher in pwCF and that inversely correlated with lung function. LasR⁻ strains showed improved growth on the majority of these compounds, many of which were amino acids, and epistasis analysis confirmed that the improved growth was due to altered activity of the CbrB-CrcZ-Crc pathway.

## Results
### Mathematical model built from monoculture growth data predicts the observed rise of *lasR* loss-of-function mutants

Our previous work on microbial interactions involving LasR⁺ and LasR⁻ *P. aeruginosa* revealed subtle differences in growth kinetics (*Mould et al., 2020*). In monoculture, *P. aeruginosa* strain PA14 Δ*lasR* had no lag phase, while the wild type had a lag phase of 1 h (*Figure 1A* for summary data and *Figure 1—figure supplement 1* for growth curve). Furthermore, consistent with work by others, the Δ*lasR* strain had a lower growth rate (*García-Contreras et al., 2020*) but a 1.5-fold higher yield in LB (*Diggle et al., 2007*). We found no differences in death rate resulting from elevated culture pH (as has previously been reported in low oxygen conditions [*Heurlier et al., 2005*]) or the onset of death phase relative to PA14 wild type under these conditions (*Figure 1—figure supplement 1*).

We built a mathematical model of strain competition exclusively from experimentally determined monoculture growth parameters to predict the relative changes in wild type and LasR⁻ cell numbers when grown on a common pool of growth substrates in order to determine how differences in growth kinetics alone would impact the rise of LasR⁻ lineages (*Figure 1A*). We modeled cell density (*Figure 1A* -left y-axis) and the percentage of LasR⁻ cells (*Figure 1A* -right y-axis) assuming a shared nutrient source and a passage every 48 hr which is a regime used previously to study the selection for LasR⁻ cells (*Heurlier et al., 2005*). Based on the mutation frequency of the *P. aeruginosa* strain PA14 ($0.52 \times 10^{-3}$ per genome per generation) (*Dettman et al., 2016*) and the size of the *lasR* gene (720 bp) relative to the genome (~ 6 Mbp), we approximate 50 *lasR* alleles with nucleotide changes would be present in a dense culture ( ~ $10^8$ cells), a fraction of which would lead to a LasR⁻ phenotype. With the assumption of 2 to 20 LasR⁻ cells per inoculum (t = 0, ~ $10^5$ cells), the model predicted that ~ 20% of the population would consist of LasR⁻ cells by Day 4, with increased percentages of ~ 40% and ~ 80% by Days 6 and 8, respectively (*Figure 1A*). Only minor differences in percentages resulted from changes in the initial LasR⁻ population.

We compared the model output to experimental data gathered with the same growth conditions and evolution regime. A single PA14 wild-type colony was used to inoculate a 5 mL culture of LB, which was grown to saturation and then used to inoculate three 5 mL LB cultures which were then passaged independently. Results from all three replicates from four independent experiments are shown. The percent of cells with LOF phenotypes were enumerated by plating and determining the percent of colonies with the characteristic 'sheen' colony morphology of LasR⁻ cells that result from accumulation of 4-hydroxy-2-heptylquinoline (HHQ) (*Figure 1B*; *D'Argenio et al., 2007*). In all four independent experiments, the percentage of colonies with the LasR⁻ phenotype rose from undetectable levels to an average of ~ 80% over the course of 8 days (*Figure 1B*). To validate the use of colony sheen as an indicator of the LasR⁻ genotype, we evaluated ≥ 90 isolates with the characteristic LasR⁻ colony morphology for other phenotypes associated with LOF: low production of proteases and autoinducers (3OC12HSL and C4HSL). Most of the predicted LasR⁻ isolates (~ 90%) had phenotypes

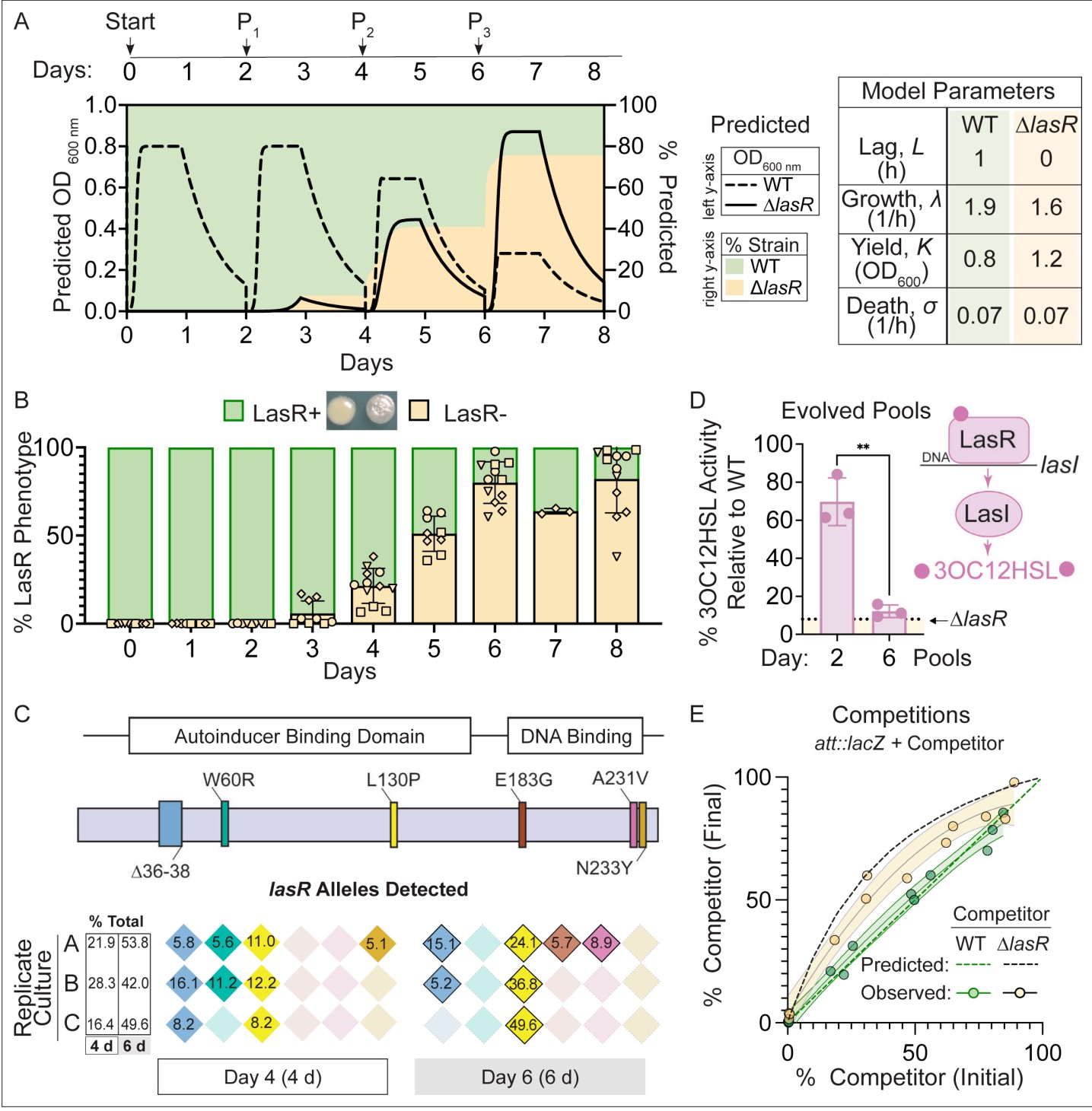

**Figure 1.** Mathematical model built from monoculture growth data is sufficient to explain the rise of LasR loss-of-function strains. (**A**) Predicted densities (left y-axis) of mathematical model shown for wild type (WT, dashed line) and LasR⁻ (solid line) strains. Predicted percentages (right y-axis) of LasR⁻ (beige fill) and LasR+ (green fill) strains over the course of evolution regime in LB with passage (P_n) every 2 days. Table shows experimentally measured growth parameters used to create the model as obtained for strains PA14 WT and Δ*lasR* grown in 5 mL LB cultures on a roller drum. (**B**) Percentage of LasR⁻ phenotypes observed in n ≥ 4 independent evolution experiments in LB. Different shapes represent independent experiments, and same shapes represent technical replicates. A representative image of the smooth LasR+ and sheen LasR⁻ colonies from Day 6 is shown. (**C**) *lasR* alleles detected in the population at Day 4 (4 d) and Day 6 (6 d) by PoolSeq within the *lasR* coding sequence, which includes the autoinducer binding and DNA binding domains, for a representative experiment (diamond symbols, in B). The percentages of each allele and the sum (i.e. % total) is indicated for each replicate culture. Each color represents a different allele. (**D**) LasR regulates the production of its cognate autoinducer 3OC12HSL via direct

*Figure 1 continued on next page*

*Figure 1 continued*

transcriptional control of the gene encoding the LasI synthase. LasI-produced autoinducer activity of evolved pools from a representative experiment (diamond symbols, in B) at days 2 and 6. Activity is presented as a percentage of that produced by unevolved WT monocultures. The levels produced by the engineered Δ*lasR* control strain is shown for reference (dotted line). **, p = 0.0015 as determined by two-tailed, unpaired t-test. (**E**) Comparison of predicted (dashed line) and observed (solid line) outcomes of competition assays initiated at different initial ratios for which a constitutively tagged WT (*att::lacZ*) was competed against Δ*lasR* (beige, gray line) or WT (control, green) competitors for 6 h (final) in planktonic LB cultures with 95% confidence intervals shown for best fit line (quadratic).

The online version of this article includes the following figure supplement(s) for figure 1:

**Figure supplement 1.** Experimentally determined growth parameters of PA14 wild type and Δ*lasR* monocultures in LB.

**Figure supplement 2.** Phenotype analysis of sheen LasR⁻ candidates isolated from the evolution experiments in LB.

that mirrored those of the PA14 Δ*lasR* strain, and not wild type (*Figure 1—figure supplement 2*). Consistent with other studies (*Feltner et al., 2016*), approximately 15% of the cells with other LasR⁻ phenotypes produced high levels of C4HSL even though 3OC12HSL production was low.

The percentage of LasR⁻ cells predicted by the model matched the frequency of *lasR* alleles in genome sequence data from pools of colonies obtained from Day 4 and 6 cultures of a representative experiment (diamond symbols in *Figure 1B*). Across replicates, six non-synonymous mutations were identified in *lasR* in the regions corresponding to the LasR autoinducer binding (Δ36–38, W60R, and L130P) and DNA binding domains (E183G, A231V, and N233Y) (*Figure 1C* and *Supplementary file 1*), which are important for function (*Feltner et al., 2016*). No synonymous mutations in *lasR* were detected. Two mutations (Δ36–38 and L130P) were present in all three replicate cultures at Day 4 and thus were likely present in the initial inoculum. In replicate A, two additional mutations in *lasR* (E183G and A231V) were identified at Day 6; the LasR A231V substitution has been extensively characterized as a loss-of-function mutation through phenotyping and genetic complementation (*Luján et al., 2007*; *Qi et al., 2016*). The percentage of *lasR* mutants in the evolved population detected by sequencing at Day 4 (22.2 ± 6.0% s.d.) and Day 6 (48.5 ± 4.9% s.d.) (*Figure 1C*) closely resembled the percentage of LasR⁻ strains predicted by the model (~ 20% and ~ 50%, respectively) (*Figure 1A*). The increased frequency of cells with the allele encoding the L130P substitution (*McCready et al., 2019*) between Day 4 and Day 6, with 13.1%, 24.6%, and 41.4% increases in replicate cultures, suggests strong selection for this particular variant or the presence of an additional mutation(s) in this background. In support of the significant increases in LasR⁻ subpopulations, the evolved cultures themselves had lower levels of the LasR-regulated autoinducer 3OC12HSL; by Day 2, culture 3OC12HSL levels were ~ 30% lower than a non-evolved wild-type culture, and showed a ~ 90% reduction by Day 6 (*Figure 1D*).

To further test the predictive power of our model for the rise of LasR⁻ lineages, we initiated cultures with different ratios of a constitutively tagged wild type (*att::lacZ*) against untagged wild-type or Δ*lasR* mutant competitors. A control assay demonstrated that the ratios of tagged and untagged wild type were unchanged over the course of growth, as indicated previously under distinct conditions (*Clay et al., 2020*; *Mould et al., 2020*). When the Δ*lasR* competitor was cultured with the tagged wild type for 6 hr, the percentage of Δ*lasR* mutant cells in the total population increased regardless of the initial percentage of Δ*lasR* (1–85%) at the time of inoculation (*Figure 1E*). The model successfully predicted that Δ*lasR* would outcompete the wild type over this range which is consistent with differential growth kinetics playing a major role (*Figure 1E* -dotted line). No Δ*lasR* advantage would be observed when it is at high initial percentages if its advantage was solely due to exploitation of common goods, as is observed when WT and Δ*lasR* are co-cultured on a substrate that requires WT protease production (*Sandoz et al., 2007*). There were differences between the best fit lines for the actual and predicted data that could be due to a variety of factors including measurement error or biological interactions between WT and Δ*lasR* strains (e.g policing *Castañeda-Tamez et al., 2018*; *Wang et al., 2015*).

## Activity of CbrAB, the two-component system that regulates carbon utilization, is required for the rise of LasR⁻ strains

To test which genes or pathways were required to promote the selection of LasR⁻ cells, we applied reverse genetics to experimental evolution. In *P. aeruginosa,* the sensor kinases of two-component systems, encoded throughout the genome, respond to a variety of diverse internal and environmental cues, such as nutrient limitation or stresses, that may be relevant to differential fitness (*Rodrigue*

*et al., 2000*; *Wang et al., 2021*). Using a library of 63 sensor kinase deletion mutants (*Wang et al., 2021*), we screened each mutant for the rise of LasR⁻ phenotypes in triplicate in a 96-well plate format (*Figure 2—figure supplement 1*). In the primary microtiter dish-based screen, in which the investigators were blind to mutant strain identity, five gene knock-outs (Δ*cbrA*, Δ*gacS*, Δ*fleS*, ΔPA14_64580, and ΔPA14_10770) showed no detectable 'sheen' colony phenotypes characteristic of LasR⁻ strains in any of the three replicates (*Figure 2—figure supplement 1* & *Supplementary file 2*). In a secondary screen of these five mutants in five mL cultures, only the Δ*cbrA* mutant (*Figure 2A*) did not evolve LasR⁻ phenotypes after serial passage; the other four mutants all had significant subpopulations with LasR⁻ phenotypes by Day 6 (*Figure 2—figure supplement 2A*). LasR⁻ strains rose with a similar frequency as in the wild type progenitor when evolution experiments were initiated with strains lacking the regulator Anr, important for LasR⁻ microoxic fitness, or the regulator RhlR, important for *lasR* mutant policing (*Chen et al., 2019*; *Clay et al., 2020*) suggesting that these regulators were not major contributors to fitness under these conditions (*Figure 2—figure supplement 2B*).

CbrA, through its regulation of the response regulator CbrB, (*D'Argenio et al., 2007*; *Sonnleitner et al., 2009*), controls *P. aeruginosa* preferential catabolism of certain carbon sources, such as succinate, over others (e.g. amino acids) through a process referred to as catabolite repression. In support of the finding that CbrA was essential for the evolution of LasR⁻ lineages, the Δ*cbrB* mutant also showed a striking and significant reduction in LasR⁻ phenotypes over the course of 8 days (*Figure 2A*). Additionally, evolution experiments in a LasR⁺ cystic fibrosis clinical isolate (DH2417) showed a similar rise in LasR⁻ phenotypes over the course of evolution, which was delayed and reduced in a Δ*cbrB* derivative (*Figure 2A*). CbrAB-controlled catabolite repression is regulated by Crc, in complex with the RNA-binding protein Hfq, which together repress the translation of target mRNAs involved in the transport and catabolism of less preferred substrates (*Figure 2B*; *Sonnleitner et al., 2017*). Crc activity is down regulated by the small RNA *crcZ*, which sequesters Crc away from its mRNA targets. The CbrAB two-component system transcriptionally regulates levels of *crcZ* (*Figure 2B*; *Sonnleitner et al., 2009*) in response to signals that have yet to be described.

Consistent with the absence of LasR⁻ phenotypes in evolved Δ*cbrA* or Δ*cbrB* cultures, Pool-Seq analysis found no mutations in *lasR* on either Day 4 or 6 (*Figure 2C*, pink and *Supplementary file 1*) which was in striking contrast to the multiple LasR⁻ alleles observed in wild type cultures. The absence of *lasR* mutations in the Δ*cbrA* and Δ*cbrB* derivatives was not due to differences in mutation frequency or number of generations as other mutations in distinct pathways under selection (e.g. *fleR* in *Figure 2C*) were present at comparable levels in all cultures (*Supplementary file 1* for data). In addition, strain PA14 wild type and the Δ*cbrA* mutant had similar growth patterns as assessed by daily optical density measurements (*Figure 2—figure supplement 2C*). We also assessed a number of factors other than differential growth that could affect the rise of LasR⁻ lineages. A previous report *Heurlier et al., 2005* found that LasR⁻ strains in the PAO1 background undergo less severe alkaline-induced lysis in another complex medium (nutrient yeast broth) when grown aerobically, but we found no evidence of differential lysis in LB between wild-type and Δ*lasR* strains under our conditions (*Figure 1—figure supplement 1A*). Furthermore, buffering the medium to pH 7 suppressed medium alkalinization (from pH of 6.8–8.5) and lysis (*Crocker et al., 2019*), but not the rise of LasR⁻ lineages; though, the kinetics of LasR⁻ lineage detection was delayed with buffering (*Figure 2—figure supplement 2D* and *Sandoz et al., 2007*). Lastly, to assess potential differences in toxicity of the wild type and Δ*cbrB* mutant culture supernatants toward LasR⁻ cells through the production of secreted factors (*Yan et al., 2018*), we grew the Δ*lasR* mutant in spent filtrate from wild-type and Δ*cbrB* cultures; no significant differences in colony forming units were observed (*Figure 2—figure supplement 2E*).

The activation of CbrAB increases growth on diverse metabolites by inducing *crcZ* which sequesters Crc away from the targets that it translationally represses (*Figure 2B* for pathway). In D'Argenio et al. (*D'Argenio et al., 2007*), higher CbrB levels were observed in LasR⁻ strains in a proteomics analysis, but no direct interactions between LasR and components of CbrA-CbrB-*crcZ*-Crc pathway have been described. Because CbrA, CbrB, and *crcZ* act to repress Crc, we hypothesized that if the loss of LasR function led to higher activity of the CbrA-CbrB-*crcZ* pathway and less Crc translational repression, we might also observe loss-of-function mutations in the genes encoding Crc or Hfq in the absence of *cbrB* (*Figure 2B*). Interestingly, the pooled genome sequence data from the Day 4 (open symbols) and Day 6 (grey symbols) populations evolved in the Δ*cbrA* and Δ*cbrB* backgrounds identified seven different mutations in *crc*, including three nonsense mutations, four missense mutations,

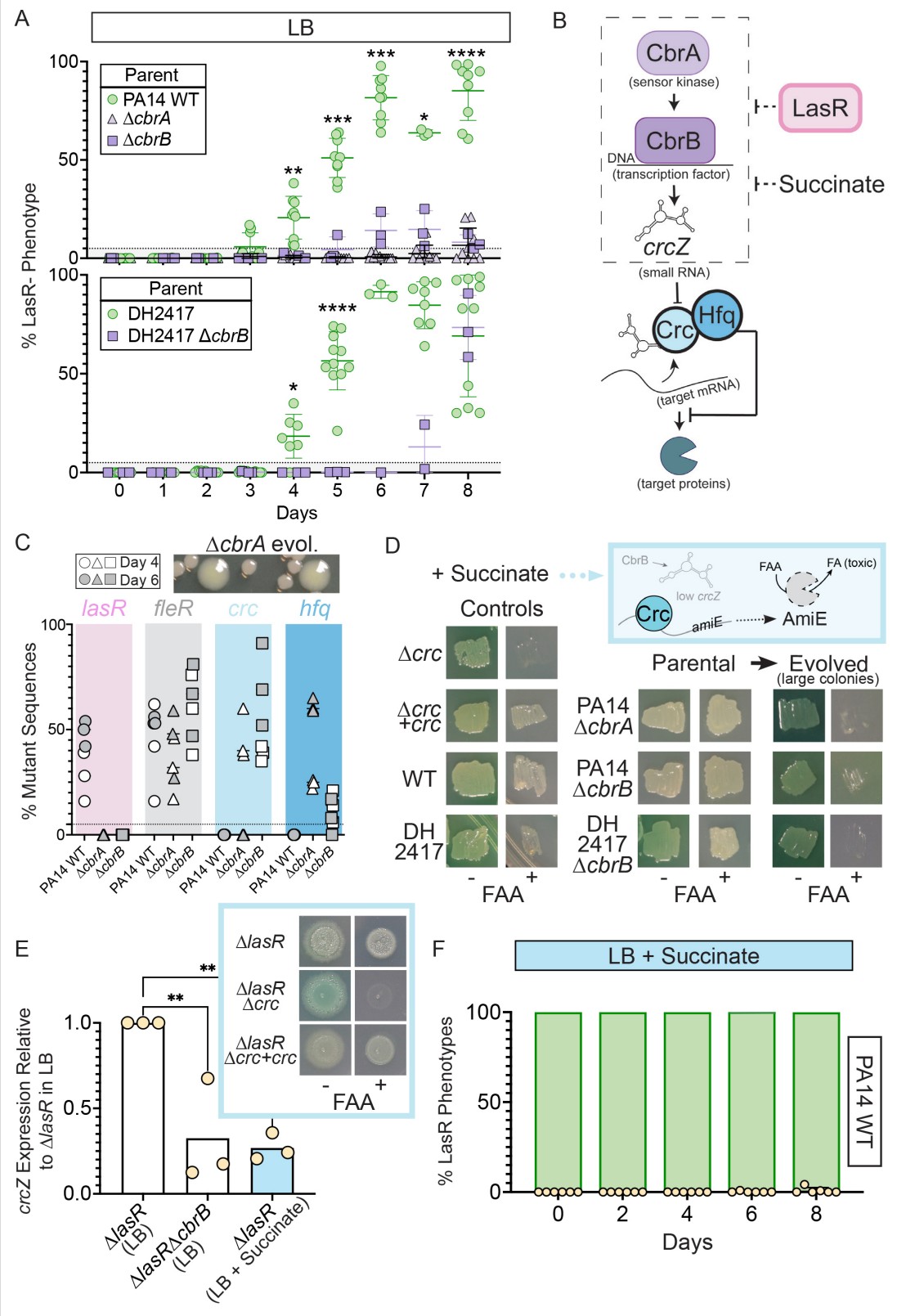

**Figure 2.** Activity of the carbon catabolite repression system is required for LasR⁻ selection in LB. (**A**) The percentage of colonies with LasR⁻ phenotypes enumerated over the course of evolution for ΔcbrA or ΔcbrB mutants (purple triangle and square, respectively) in strains PA14 or a LasR+ cystic fibrosis isolate (DH2417) relative to 'wild-type' comparators. PA14 WT strain data is the same as in *Figure 1B* (n ≥ 3). Statistical significance was determined between percent LasR⁻ phenotypes in CbrA/B + and *cbrA/B* mutant pools each day via Two-Way ANOVA with Dunnet's multiple hypothesis correction.

*Figure 2 continued on next page*

Figure 2 continued

For all panels: *, p < 0.05; **, p < 0.005; ***, p < 0.0005; ****, p < 0.0001. (**B**) The carbon catabolite repression system promotes the preferential consumption of succinate (and other preferred substrates) through the two-component system CbrAB. CbrA activates its response regulator CbrB which directly induces expression of the small RNA *crcZ*. *crcZ* sequesters Crc thereby allowing translation of the target gene to occur. Often the target gene enables the utilization of specific (i.e. less preferred) substrates. In a catabolite repressed state, such as when succinate is present, Crc binds to target mRNA with the RNA binding protein Hfq and blocks translation. CbrB protein levels are higher in strains lacking LasR function, but the mechanism linking these pathways is uncharacterized. (**C**) Percent total mutant alleles in *lasR* (pink bar), *fleR* (gray bar), *crc* (light blue bar), and *hfq* (darker blue bar) in a representative experiment (**Figure 1B**, diamond symbols) for PA14 wild type, ΔcbrA, ΔcbrB evolved populations sequenced on Day 4 (white filled symbol) and six (gray filled symbol). Representative image of the larger colony morphologies observed in the evolved pools from CbrA/B- deficient strains (ΔcbrA shown) above. (**D**) Crc represses *amiE* encoding an amidase that can turnover the fluoroacetamide (FAA) protoxin to fluoroacetate (FA) mediating cell death. In the presence of succinate, cells with functional Crc survive in the presence of FAA. PA14 WT, PA14 ΔcbrA, PA14 ΔcbrB, and DH2417 WT strains were included as controls. The ΔcbrA and ΔcbrB parental strains used for the evolution experiments and representative colonies that emerged with a larger colony size in these backgrounds were patched (or struck out) onto succinate containing plates in the absence and presence of the FAA protoxin. (**E**) *crcZ* expression of PA14 ΔlasR in LB (white bar) and LB with 40 mM succinate (blue bar) measured by qRT-PCR and plotted relative to expression of ΔlasR in LB (n = 4). Inset shows representative image of ΔlasR, ΔlasRΔcrc, and ΔlasRΔcrc +crc grown on succinate containing plates in the absence and presence of FAA. (**F**) Percentage of colonies with LasR⁻ phenotypes observed in evolution experiment initiated with strain PA14 WT in LB supplemented with 40 mM succinate (n = 6). P-values of 0.007 and 0.005 for comparison of ΔlasRΔcbrB (LB) and ΔlasR (LB +succinate) relative to ΔlasR grown in LB as determined by ordinary one-way ANOVA with Dunnett's multiple comparison test.

The online version of this article includes the following figure supplement(s) for figure 2:

**Figure supplement 1.** Screen reveals specific requirement of CbrA, and not other sensor kinases encoded in the PA14 genome for LasR⁻ strain selection.

**Figure supplement 2.** LasR⁻ strains evolve in all tested strain backgrounds except for those deficient in *cbrAB,* and this is independent of cellular density, lysis, and filtrate toxicity.

and six indels, and these were among the most abundant mutations in the ΔcbrB mutant cultures; no *crc* mutations were identified in the PA14 wild type evolved populations (**Figure 2C**). In ΔcbrB, *crc* mutant alleles showed the largest rise between Day 4 and Day 6 across all three replicate cultures (**Supplementary file 1**). In the ΔcbrA passaged cultures, we also identified a rise in *hfq* mutations within the coding and upstream intergenic regions (**Figure 2C** and **Supplementary file 1** for sequence data) in addition to mutations in *crc*. The changes in relative abundances of alleles with mutations in *crc* and either the promoter or coding regions of *hfq* across the 2 days suggested that *hfq* mutations and *crc* mutations were in different backgrounds (**Supplementary file 1**).

To assess Crc-Hfq function in evolved strains, we leveraged Crc translational repression of the amidase AmiE, which cleaves the prototoxin FAA to the toxic FA (**Figure 2D** for pathway) (**O'Toole et al., 2000**). Succinate, which downregulates CbrAB activity, maintains repression of AmiE, thereby enabling wild type to grow in the presence of FAA. In the absence of functional Crc or its co-repressor Hfq, cells synthesize AmiE, and FAA conversion into FA inhibits growth. As expected, on medium with succinate, FAA inhibited growth of the Δcrc mutant, but did not affect growth of the complemented Δcrc +crc strain, the wild type, and the ΔcbrA and ΔcbrB mutants. However, in passaged ΔcbrA and ΔcbrB cultures, spontaneous mutants in the population gave rise to larger colonies (**Figure 2C**, top), and these isolates were FAA sensitive (**Figure 2D**) supporting the model that in the ΔcbrA and ΔcbrB backgrounds, mutations that abolished Crc or Hfq activity arose. Secondary mutants with FAA sensitivity also arose in the DH2417 ΔcbrB background upon passaging, indicating that this phenomenon was not unique to the PA14 background, and another study also reported *crc* and *hfq* mutants in the absence of *cbrB* (**Boyle et al., 2017**). Given the apparent selection for decreased Crc function in ΔcbrA and ΔcbrB, and the requirement of *cbrA* or *cbrB* for LasR⁻ strain selection, we hypothesized that increased CbrAB activity may be a trait that increases the fitness of LasR⁻ strains.

To complement the genetics approach of evolution assays in *cbrAB* mutants, we monitored the rise of LasR⁻ lineages in LB medium supplemented with succinate, which inhibits CbrAB activity (**Sonnleitner et al., 2009**). Medium amendment with 40 mM (pH 7) succinate was sufficient to repress CbrB-regulated *crcZ* small RNA expression in ΔlasR to levels reminiscent of ΔlasRΔcbrB (**Figure 2D**). *lasR* mutants still responded to succinate; succinate reduced *crcZ* levels in ΔlasR and enabled ΔlasR growth on medium with FAA due to Crc activity (**Figure 2E**, inset). This indicated that ΔlasR retains the Crc-Hfq mediated translational repression when succinate is present. Succinate amendment suppressed the rise of LasR⁻ phenotypes in PA14 wild type (**Figure 2F**).

## Elevated *cbrB* and *crcZ* expression and reduced Crc-dependent repression are sufficient to recapitulate the growth advantages of LasR⁻ strains

CbrAB activity induces the expression of *crcZ*, which sequesters Crc. We found that the Δ*lasR* mutant had ~two fold higher *crcZ* levels compared to wild type, suggesting higher activity of the CbrAB two-component system in LasR⁻ strains (*Figure 3A*). Previous work reported higher yields on phenylalanine for LasR⁻ relative to LasR⁺ strains concomitant with elevated CbrB protein levels in a proteomics analysis (*D'Argenio et al., 2007*). Thus, we first used phenylalanine as a growth substrate to further dissect the activity of the CbrAB-*crcZ*-Crc pathway (*Figure 2B* for pathway) in LasR⁻ strains. In planktonic cultures in medium with phenylalanine as a sole carbon source, the Δ*lasR* strain obtained significantly reduced lag (*Figure 3B*) and higher yields (*Figure 3C*) than the wild type, and the enhanced growth phenotype was complementable by *lasR*. As previously reported, growth on phenylalanine depended on *cbrB*; the Δ*cbrB* and Δ*lasR*Δ*cbrB* mutants grew similarly poorly and their growth yield could be fully complemented by expressing *cbrB* (*Figure 3B and C*). Deletion of *crc* in the Δ*lasR*Δ*cbrB* strain also restored growth to levels comparable to the Δ*lasR* and Δ*lasR*Δ*cbrB*+*cbrB* strains (*Figure 3B and C*) indicating Crc repression of phenylalanine catabolism in the *cbrB* mutant. Overexpression of either *cbrB* or its target *crcZ*, which acts as a Crc-sequestering agent, was sufficient to improve yields on phenylalanine relative to the empty vector control (*Figure 3D*).

The CbrB- and Crc-controlled growth advantage on phenylalanine for LasR⁻ strains in planktonic cultures was also apparent in colony biofilms (*Figure 3—figure supplement 1A*). In colony biofilms, again, Δ*lasR* had improved growth on phenylalanine, that was dependent on *cbrB* and the growth defect of the *lasRcbrB* mutant could be rescued by deletion of *crc* (*Figure 3—figure supplement 1A*). The same pattern was observed on other substrates for which catabolism is under the control CbrAB-Crc pathway such as glucose and mannitol (*Figure 3—figure supplement 1A*). While deletion of *crc* was able to restore enhanced growth to the Δ*lasR*Δ*cbrB* mutant, Δ*crc* did not grow as robustly as the Δ*lasR* mutant which is consistent with the detection of LasR⁻ lineages but not Crc⁻ lineages in passaged wild type cultures. Thus, LasR⁻ strains from stationary phase cultures appear to be primed for growth on multiple single carbon sources under CbrB-Crc control and reach higher final yields on these substrates.

## LasR⁻ strains have CbrB-dependent growth advantages on metabolites enriched in progressive cystic fibrosis lung infections

Loss-of-function mutations in *lasR* are commonly detected in samples from chronic *P. aeruginosa* lung infections in pwCF, and these mutants have been correlated with a more rapid rate in lung function decline (*Hoffman et al., 2009*). To determine the metabolite milieu in the CF lung, we performed a metabolomics analysis of bronchioalveolar lavage samples collected from 10 pwCF and 10 non-CF individuals (*Supplementary file 3*). The pwCF were infected with diverse pathogens and had varying lung function, which was measured as forced expiratory volume in 1 s and presented as the percent expected at one's age (%FEV$_1$). Over 300 compounds were measured, and no uniquely microbial metabolites were noted. Many compounds were higher in the CF population, but some were unchanged (e.g glucose) and others were higher in non-CF samples (e.g adenosine and glutathione as previously published *Esther et al., 2008*; *Fitzpatrick et al., 2014*; *Supplementary file 4*).

In a principal component analysis (PCA), samples from non-CF individuals clustered together while those from pwCF were more spread. Samples from pwCF with high lung function (112 or 113 %FEV$_1$) grouped among the non-CF samples (*Figure 4A*). The metabolites that contributed strongly to the first principal component, PC1, showed a significant inverse correlation with %FEV$_1$ including phenylalanine, arginine, lactate, and citrate (*Figure 4B*). As with phenylalanine (*Figure 3B and C* & *Figure 3—figure supplement 1A*), the Δ*lasR* strain had growth advantages on arginine, lactate, and citrate that were controlled by CbrB and Crc (*Figure 3—figure supplement 1A, B*).

We identified the 20 carbon sources that were most enriched in CF samples including those that correlated inversely with lung function, then used a BIOLOG phenotype array to assess whether the trend of greater yield for the Δ*lasR* strain persisted across this set. We found a significantly higher yield for the Δ*lasR* strain when we analyzed growth of the WT and Δ*lasR* across this group of 20 carbon sources suggesting that Δ*lasR* has improved growth on many of the nutrients available in the lung (*Figure 4C*).

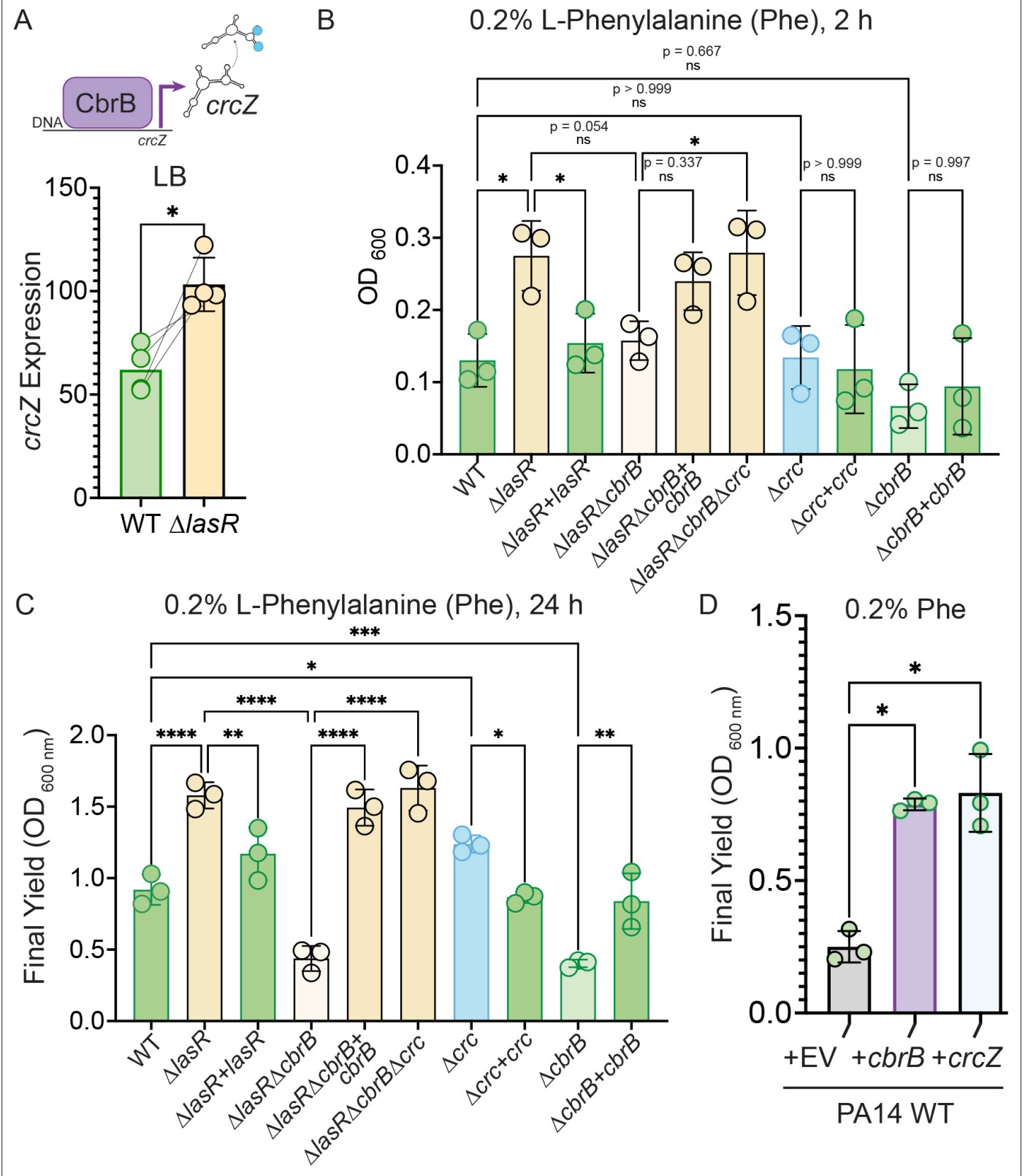

**Figure 3.** Increased CbrB activity of LasR⁻ strains is necessary and sufficient to promote growth on non-repressive substrates like phenylalanine via Crc. (**A**) CbrB promotes the transcription of *crcZ*, and *crcZ* thus can be a readout of CbrB transcriptional activity. *crcZ* expression was measured by qRT-PCR relative to the average expression of the housekeeping genes *rpoD* and *rpsL* in cultures of PA14 WT and Δ*lasR* strains grown to $OD_{600\,nm} = 1$ from four independent experiments. *, p = 0.0334 as determined by Student's paired two-tailed t-test. (**B**) Early growth (2 hr) and (**C**) Final yield (24 hr) on

*Figure 3 continued on next page*

*Figure 3 continued*

phenylalanine (Phe) as a sole carbon source shows enhanced growth for Δ*lasR*, *cbrB* dependence, and the requirement for *cbrB* is abolished by deletion of *crc*. Each point is the average of three replicates, repeated three independent days. Statistical significance determined by one-way ANOVA with Šídák's multiple comparisons test. ns, not significant. *, $p < 0.05$. **, $p < 0.005$. ****, $p < 0.0001$. (**D**) Final yield on Phe under (0.2%) arabinose-inducing conditions for the PA14 WT strain expressing an empty vector or *crcZ*, and *cbrB* overexpression constructs. Each point is the average of three replicates, performed on three separate days. Statistical significance determined by one-way ANOVA with multiple hypothesis correction as above.

The online version of this article includes the following figure supplement(s) for figure 3:

**Figure supplement 1.** Δ*lasR* strains have CbrB-dependent growth advantages that can be restored via loss of *crc*.

To further test the hypothesis that the growth phenotypes of LasR⁻ strains can promote selection in the nutrient environment of the CF lung, we performed evolution experiments using both strain PA14 and a LasR⁺ CF clinical isolate in a medium designed to more closely recapitulate the nutritional profile of the cystic fibrosis airway. Upon absolute quantitation, we observed good concordance between the relative abundances of amino acids found in BAL fluid and reported for sputum (*Palmer et al., 2005*) which served as a basis for an artificial sputum medium, ASM (*Figure 4—figure supplement 1*; *Clay et al., 2020*) that was based on a previously reported synthetic CF medium (SCFM2) (*Palmer et al., 2005*). LasR⁻ strains evolved in both strain backgrounds (*Figure 4D*) with kinetics similar to what was observed in LB medium (*Figure 1B*). Parallel evolution experiments in ASM initiated with Δ*cbrB* derivatives did not exhibit a rise in LasR⁻ phenotypes in either strain background to suggest that CbrAB activity was again a contributor to the fitness of *lasR* LOF mutants.

## Discussion

Through mathematical modeling, experimental evolution, and competition assays, we found that the rise of problematic *P. aeruginosa* LasR⁻ variants frequently observed in disease could be explained by increases in yield and decreases in lag during growth on carbon sources abundant in the lung environment (*Figure 5*). In fact, the steady state growth rate for Δ*lasR* was slightly less than that for the wild type, which is consistent with the model that there are frequently tradeoffs between a shorter lag phase and overall growth rate (*Basan et al., 2020*). Interestingly, CF-adapted *P. aeruginosa* isolates have been found to have slower in vitro growth rates than other strains (*Yang et al., 2008*). Other factors will impact the relative fitness of LasR⁺ and LasR⁻ cells across different growth phases (*Figure 5*) including oxygen availability and pH buffering capacity, which may lead to differential lysis (*Heurlier et al., 2005*), or the need for (or exploitation of) proteases to gain access to growth substrates (*Van Delden et al., 1998*; *Sandoz et al., 2007*).

The overlap between the model-predicted and observed percentages of LasR⁻ strains over the course of the evolution regime suggests that the described social advantage resulting from QS dysfunction for LasR⁻ strains (i.e. social cheating) is not necessary to explain the timing and kinetics of the initial rise of LasR⁻ strains under our conditions. However, social interactions that benefit LasR⁻ strains may be evident where the model estimates percentages that fall below or on the lower range of that experimentally observed, such as Day 4 or 6. A more detailed discussion of when social interactions are required is found below.

The data presented support the model that that increased growth of LasR⁻ cells on many amino acids, sugars, and lactate is due to higher CbrAB-controlled *crcZ* levels which downregulates metabolism under Crc control, and these findings nicely parallel studies by D'Argenio et al. (*D'Argenio et al., 2007*) that found higher levels of CbrB in LasR⁻ isolates. In PA14 Δ*cbrA* and Δ*cbrB* mutants, *lasR* LOF mutations did not arise, but mutations in *crc* and upstream of *hfq* were observed. As *crc* mutations phenocopy some of the growth advantages of the *lasR* mutants (*Figures 2C and 3*, *Figure 3—figure supplement 1*), the importance of derepressed catabolism for fitness is underscored. It is interesting to note that there were differences in the relative dependence on CbrB for the selection for LasR⁻ between strains PA14 and DH2417, and in ASM the dependence on *cbrB* for the selection of LasR⁻ strains increased in both strains (*Figure 4D*) suggesting that different environments may alter the importance of different LasR- and CbrB-controlled targets important for fitness that have yet to be elucidated. Though deletion of *cbrA* or *cbrB* can have pleiotropic effects (*Yeung et al., 2011*), we did not observe differences in density, quorum sensing regulation, production of quorum sensing controlled factors such as proteases, lysis in stationary phase, or overall mutation accumulation

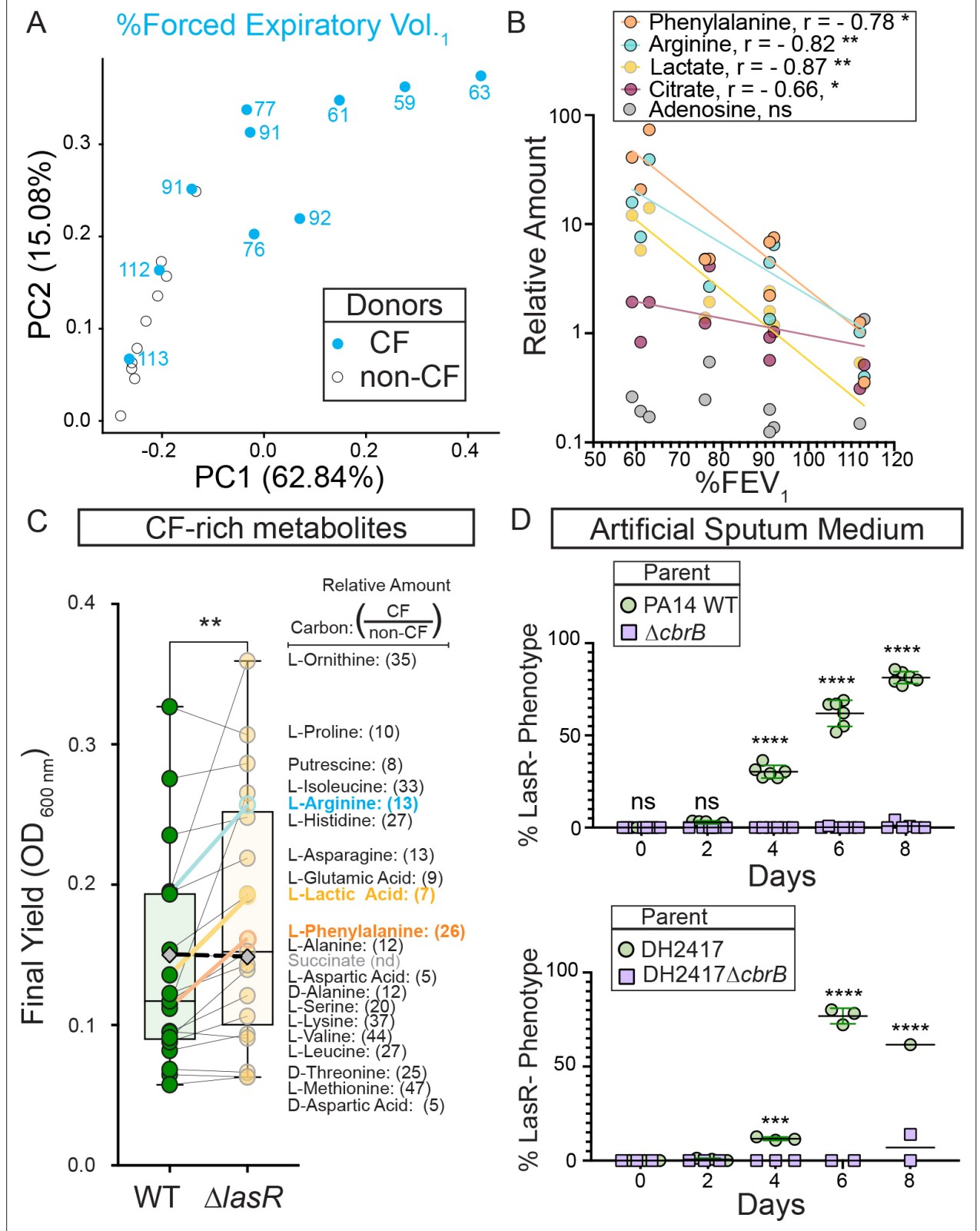

**Figure 4.** CbrB-dependent growth advantages may contribute to *lasR* mutant selection in distinct nutrient profiles of progressive cystic fibrosis airways. (**A**) The first two dimensions (PC1 and PC2) of a principal component analysis of log normalized metabolite counts from bronchoalveolar lavage (BAL) samples collected from cystic fibrosis (CF, blue filled) and non-cystic fibrosis (non-CF, gray open) donors explain 62.84% and 15.08% of the variation in the data, respectively. PC1 separates the metabolite data by relative lung function as measured by percent forced expiratory volume in 1 s (%FEV$_1$) for

*Figure 4 continued on next page*

*Figure 4 continued*

samples from people with CF. The %FEV₁ is overlayed for CF-donor samples with text. Samples from non-CF donors group more closely with CF-donor samples that have high lung function. (**B**) Spearman correlation analysis of the relative phenylalanine (orange), arginine (aqua), lactate (yellow), citrate (magenta), and adenosine (gray) metabolite counts in the BAL samples relative to %FEV₁ with exact p-values of 0.010, 0.005, 0.002, 0.041, and 0.714, respectively. Best fit semilog lines shown for visual clarity. (**C**) Comparison of the final yield measured after 24 hr for strains PA14 wild type and Δ*lasR* across a subset of carbon sources in BIOLOG growth assays for which the metabolite was found to be in higher abundance in CF-donor relative to non-CF donor BAL samples. Bold font indicates carbon sources analyzed in *Figure 3* and *Figure 3—figure supplement 1*. Number in parenthesis refers to the ratio of the average counts for each metabolite in CF relative to non-CF samples. **, p-value = 0.003 as determined by two-tailed paired t-test comparing growth across CF-enriched metabolites between the wild type and Δ*lasR* strains. Succinate (gray diamond, black dashed line) was not detected (nd) in the BAL samples and thus not included in the statistical analysis, but the growth data is shown for reference. (**D**) Observed percentage of colonies with LasR⁻ phenotypes over the course of evolution from strains (top) PA14 WT or (bottom) CF isolate (both green circles) with Δ*cbrB* (purple squares) derivatives in artificial sputum medium (ASM), which was designed to recapitulate the CF lung nutritional profile. ns, not significant (p-value > 0.9); ***, p = 0.0008; ****, p < 0.0001 as determined by ordinary two-way ANOVA with Šídák's multiple comparisons test.

The online version of this article includes the following figure supplement(s) for figure 4:

**Figure supplement 1.** Quantitative amino acid analysis.

between wild type, Δ*cbrA*, and Δ*cbrB* that could explain differences in the rise of LasR⁻ subpopulations. Furthermore, environmental modification of CbrB activity by the addition of succinate to LB (*Sonnleitner et al., 2009*) also suppressed the emergence of LasR⁻ strains in the wild type. Because CbrAB activity can still be suppressed by succinate in LasR⁻ cells (*Figure 2E*), LasR⁻ variants were not strictly 'de-repressed', and this is consistent with the fact that Δ*lasR* and Δ*crc* growth patterns were not identical. Unlike *lasR* mutations, *crc* mutations are not commonly observed in clinical isolates (*Winstanley et al., 2016*) and *crc* mutants have been shown to be under negative selection in Tn-Seq experiments (*Lorenz et al., 2019*).

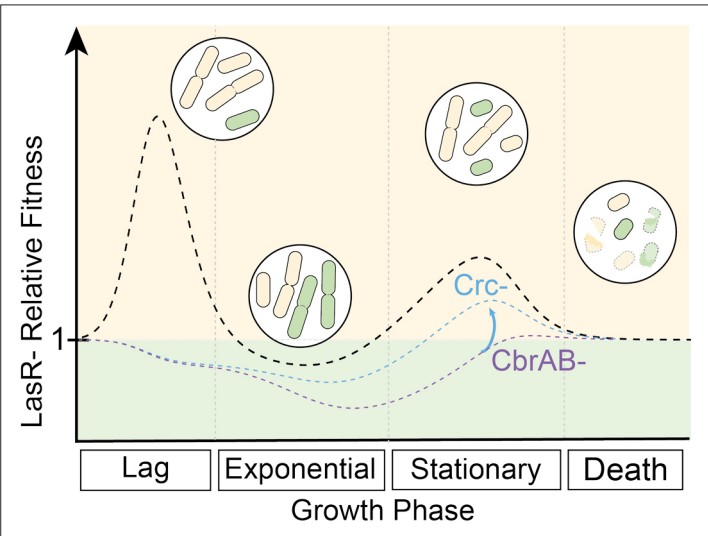

**Figure 5.** CbrAB activity contributes to the positive selection of LasR⁻ strains in complex media. LasR⁻ strain fitness relative to wild type is shown across growth phases, including lag, exponential growth, stationary, and death phases. Relative fitness of the LasR⁻ strain (dotted black line) is calculated from the experimentally determined monoculture growth data of strains PA14 wild type (WT) and Δ*lasR* over time. Values above one indicate a LasR⁻ strain fitness advantage over the WT strain during that growth phase. Circled insets show representative cartoons of LasR- (beige) and LasR+ (green) cells at each growth phase to indicate dividing or lysing cells (burst cells) across growth stages. The heights of the peaks or valleys of the relative fitness lines can be altered by several modulating factors including those that contribute to the positive and negative selection of LasR⁻ strains. Other modulating factors reported or suggested in the literature include inter- and intra- species competition, extracellular protease, immunoclearance, and oxygenation which are likely condition dependent. In the absence of CbrA or CbrB (CbrAB-, dotted purple line) or in the presence of succinate (one CbrAB repressive substrate), the relative growth of LasR⁻ strains is lower resulting in a reduction in the observed selection. This could be partially relieved in the CbrAB- background through disruption of Crc or Hfq function (blue dotted line), restoring activity through the pathway.

Analysis of BAL fluid revealed higher levels of substrates such as lactate and amino acids, which require CbrB for consumption, in samples from pwCF, and these findings are consistent with other more targeted analyses of CF airway samples (*Bensel et al., 2011*; *Twomey et al., 2013*). Consistent with our finding that higher levels of certain metabolites correlated with worse CF lung disease, other studies including that of *Esther et al., 2016* found a correlation between total metabolites and neutrophil counts suggesting host cell lysis, along with lysis of microbial cells, may be a major contributor to a shift in the metabolome. CF-lung derived *P. aeruginosa* isolates can have amino acid auxotrophies and enhanced amino acid uptake (*La Rosa et al., 2019*) which supports ready access to amino acids in vivo. Several CF isolates show reduced succinate assimilation to suggest the uptake of less preferred substrates over the course of adaptation, which may indicate decreased Crc activity over time (*Jørgensen et al., 2015*; *La Rosa et al., 2018*).

Our model predicts LasR⁻ strains benefit from growth advantages that might be present when new nutrients become available (analogous to lag phase) and in dense populations when improved yields for the Δ*lasR* mutant emerges; due to a slower steady state growth rate, we predict that LasR⁻ strains would not emerge under steady state growth conditions such as in a chemostat. Indeed, the advantages of decreased lag phase in cultures has been proposed to be a universal adaptation in dynamic environments (*Basan et al., 2020*; *Bertrand and Margolin, 2019*). Thus, the frequent emergence of LasR⁻ lineages in the CF lung and other disease settings suggests that *P. aeruginosa* often undergoes growth transitions in vivo, possibly due to fluctuating local conditions, spatial heterogeneity, or the result of complex competition between bacterial and host cell types. The CbrB-dependent rise of LasR⁻ strains in the complex CF mimetic medium (i.e. artificial sputum medium, ASM) alongside the positive selection observed in minimal media with CF relevant substrates (*Scribner et al., 2022*) shown to require CbrB for LasR⁻ strain growth enhancement suggests that the growth advantages of *lasR* mutants may be sufficient to overcome any potential negative selective pressures mediated by the host, neighboring microbes, or inaccessibility to nutrients like complex protein or adenosine. In addition, the loss of LasR function enables other inherent advantages that contribute to competitive fitness including resistance to lysis under conditions of high aeration, enhanced microoxic fitness, enhanced RhlR activity (*Chen et al., 2019*; *Clay et al., 2020*; *Heurlier et al., 2005*), and altered intraspecies interactions (*Mould et al., 2020*) which may be relevant in the complex and dynamic nutritional environment of the CF airway over the course of disease. The connection between these phenotypes and the CbrAB-*crcZ*-Crc pathway is not yet clear.

The increased growth in post-exponential phase cultures for LasR⁻ strains bears similarities to mutations that arise in other microbes. For example, the selection for *rpoS* mutants in stationary phase cultures of *E. coli* (*Finkel and Kolter, 1999*; *Zambrano et al., 1993*; *Zinser and Kolter, 2000*) is also dependent on nutrient accessibility (*Farrell and Finkel, 2003*) with enhanced amino acid catabolism as a major contributor to *E. coli* lineages with growth advantages in stationary phase (GASP) (*Zinser and Kolter, 1999*). While the rise of *rpoS* mutants in laboratory settings required pH-driven lysis (*Farrell and Finkel, 2003*), LasR⁻ strains still evolved in buffered medium suggesting distinct mechanisms for the metabolic advantages of *lasR* mutants. It is worth noting that none of the common GASP mutations (*rpoS, lrp,* or *ybeJ-gltJKL*) were identified in our in vitro evolution studies (*Supplementary file 1*). We considered that the enhanced growth of LasR⁻ strains in post-exponential growth phases may be due to differences in ppGpp signaling, given growth arrest as part of the stringent response modifies the expression of QS-regulated genes (*van Delden et al., 2001*). However, no mutations in *relA* or *spoT*, the two ppGpp synthases, were observed. The mechanism of increased CbrB activity in Δ*lasR* remains an unresolved question that is relevant to *P. aeruginosa* biology and may aid in the identification of the signals that activate the CbrA sensor kinase which influences clinically relevant phenotypes including virulence and antibiotic resistance (*Yeung et al., 2011*). Our working model is that the upregulation of CbrB transcription of *crcZ* increases levels of transporters and catabolic enzymes due to the release from Crc repression, and this enhanced substrate uptake alters intracellular metabolite pools driving metabolism in accordance with Le Chatelier's principle (*Monod, 1949*). Thus, quorum sensing mutants can maintain higher growth rates at lower substrate concentrations than for quorum-sensing intact cells.

The repeated observation that LOF mutations readily arise in diverse settings provokes the question of how quorum sensing is maintained. Several elegant mechanisms that address this point have been described. First, the wiring of the LasR regulon is such that while there are growth advantages

on many substrates present in the lung, there are growth disadvantages on other important nutrient sources (e.g. adenosine and proteins and peptides *Heurlier et al., 2005*). Social cheating can promote the rise of LOF mutants in protease-requiring environments (*Diggle et al., 2007*; *Hassett et al., 1999*). Second, there are quorum-sensing controlled 'policing' mechanisms through which LasR$^+$ strains restrict the growth of LasR$^-$ types through the release of products toxic to quorum-sensing mutants (*Castañeda-Tamez et al., 2018*; *García-Contreras et al., 2020*; *Wang et al., 2015*). Lastly, there are other tradeoffs such as sensitivity to oxidative stress that may limit LasR$^-$ lineage success (*Hassett et al., 1999*). Quorum sensing exerts metabolic control in other diverse microbes beyond *P. aeruginosa*. Thus, these data provide insight into generalizable explanations for the benefits of metabolic control in dense populations and indicate drivers for frequent loss-of-function mutations in quorum-sensing genes such as *agr* in *Staphylococcus aureus* and *hapR* in *Vibrio cholerae* (*Mould and Hogan, 2021*).

Together, these data highlight the power of coupling in vitro evolution studies with forward and reverse genetic analyses. Other benefits to this approach include the ability to dissect subtle differences between pathway components. For example, multiple mutations in *crc* repeatedly rose in Δ*cbrA*-, but not in Δ*cbrB*-derived populations, and multiple mutations in *hfq* rose in Δ*cbrB*-, and not in Δ*cbrA*-derived populations. While CbrA and B work together as do Crc and Hfq, these observations may provide a foothold into key distinctions that could yield mechanistic insights. In the future, the ability for deep sequencing of infection populations and analysis of evolutionary trajectories may aid diagnoses and treatment decisions in beneficial ways.

## Materials and methods

See Key Resources Table in supplement for additional details on key reagents.

### Strain construction and maintenance

In-frame deletions and complementation constructs were made using a *Saccharomyces cerevisiae* recombination technique described previously (*Shanks et al., 2006*). The *cbrB* and *crcZ* expression vectors were constructed by HiFi Gibson assembly with the NEBuilder HiFi DNA Assembly kit according to manufacturer's protocol. All plasmids were sequenced at the Molecular Biology Core at the Geisel School of Medicine at Dartmouth. In frame-deletion and complementation constructs were introduced into *P. aeruginosa* by conjugation via S17/lambda pir *E. coli*. Merodiploids were selected by drug resistance and double recombinants were obtained using sucrose counter-selection and genotype screening by PCR. Expression vectors were introduced into *P. aeruginosa* by electroporation and drug selection. All strains used in this study are listed in *Supplementary file 6*. Bacteria were maintained on lysogeny broth (LB) with 1.5% agar. Yeast strains for cloning were maintained on YPD (yeast extract-peptone-dextrose) with 2% agar. Artificial sputum medium (ASM) was made as described previously (*Clay et al., 2020*).

### Mathematical model

Growth parameters were determined from 5 mL grown LB cultures inoculated as described in the experimental evolution protocol and the monocarbon growth below. In brief, a 16 h overnight LB culture was normalized to OD$_{600\,nm}$ = 1 in LB, and a 250 µL aliquot of the normalized culture was used to inoculate 5 mL fresh LB for an approximate OD$_{600\,nm}$ = 0.05 at time zero. The density (OD$_{600\,nm}$) was measured for up to 48 hr by taking a 10–100 µL aliquot at the designated time intervals from the 5 mL culture tube with dilution into LB as appropriate in a 96-well plate (100 µL total per well) for OD$_{600\,nm}$ measurement using a Spectramax M2 microplate reader with Softmax Pro 6.5.1 software. Lag and growth rate were measured in separate experiments from those used to monitor lysis so that the volume in the 5 mL cultures tubes never dropped below 10% of the starting volume. See *Supplementary file 7* for additional details for the parameter choices used in the mathematical model and *Source code 1* for the Matlab script.

### Experimental evolution

Experimental evolution was modeled after work by *Heurlier et al., 2005*. A single colony of each strain was used to inoculate a 5 mL LB culture in 13 mm borosilicate tubes. The tubes inoculated with

a single colony were grown for 24 hr at 37 °C on a roller drum. The 24 hr grown culture was adjusted to $OD_{600 nm}$ = 1 in LB based on $OD_{600 nm}$ reading of a 1–10 dilution in LB of the 24 hr culture in a 1 cm cuvette using a Spectronic GENESYS 6 spectrophotometer. Separate 250 µL aliquots of the $OD_{600 nm}$ normalized cells was sub-cultured into three tubes containing 5 mL fresh media to initiate the evolution experiment (i.e. time 0) with three distinct replicate cultures per experiment. At time of passage every two days, 25 µL of culture was transferred into 5 mL fresh media. Every day (or as indicated) cultures were diluted and spread onto LB agar plates using sterile glass beads for phenotype distinction. The LB agar plates were incubated for ~ 24 hr at 37 °C and then left at room temperature for phenotype development. The sheen LasR⁻ colony morphologies were counted, and the percentage of LasR⁻ phenotypes calculated based on total CFUs. All experimental evolutions in LB were repeated on at least three independent days with three replicates of each strain per experiment unless otherwise stated. The ASM and succinate amended medium evolutions were completed on two separate days. In the case of Δ*rhlR* and Δ*anr*, the three replicates were inoculated from three independent overnights. Data visualization and statistical analysis was performed in GraphPad Prism 9 (version 9.2.0).

## gDNA extraction, sequencing, SNP calling of Pool-Seq data

Between 100 and 150 random colonies were scraped and pooled from the LB agar plates that were counted and used to measure the percent of colonies with LasR⁻ phenotypes at Days 4 and 6 from a representative WT-, Δ*crbA*-, and Δ*cbrB*-initiated evolution experiment. For plates containing a total of 100–150 colonies, all colonies on the plate were collected for a single pooled genomic DNA extraction. If more than 150 colonies were on a plate, the plate was divided equally, and all colonies in an arbitrary section were collected to ensure genomic DNA was extracted from a similar number of colonies for each sample. Scraped up cells were pelleted briefly in a 1.5 mL Eppendorf tube via a short spin, resuspended in 1 mL PBS, vortexed briefly, and gDNA was subsequently extracted from a 50 µL aliquot of cell resuspension via the Master Pure Yeast DNA purification kit according to manufacturer's protocol with RNAase treatment. A 2.5 µg aliquot was submitted for Illumina sequencing (1Gbp) at the Microbial Genome Sequencing (MiGs) Center on the NextSeq 2000 platform. The resulting forward and reverse reads were trimmed with bcl2fastq (v2.20.0422) to remove Illumina adaptor sequences during the demultiplexing process. Both forward and reverse read files were aligned and compared to the complete and annotated UCBPP-PA14 genome available on NCBI (accession GCF_000014625.1) using the variant caller BreSeq (*Deatherage and Barrick, 2014*) (version 0.35.4) with the -p option for polymorphisms and a 5% cutoff. Specifically, the following command was used: breseq -p -j 10 r [reference file] [sample name]_.fastq.gz [sample name]_fastq.gz -o [output file name]. This provided an output file that specified variations from the reference genome and listed their respective fractions of the total reads. These fractions were treated as estimations of genotype proportions in the population. Variants at fixation (100%) across all 18 samples (three strains, 2 days) were excluded from follow-up analysis as potential differences in strain background that differed from the reference genome at the start of the experiment. All sequencing data is available on the Sequence Read Archive with the accession number PRJNA786588.

## Milk proteolysis

Brain Heart Infusion Agar was supplemented with powdered skim milk dissolved in water to a final concentration of 1%. The evolved isolates selected on basis of 'sheen' colony morphology were grown in a 96-well plate with 200 µL LB per well for 16 hr. Milk plates were inoculated with ~ 5 µL of culture using a sterilized metal multiprong inoculation device (Dan-Kar) and incubated at 37 °C for 16 hr. PA14 WT and Δ*lasR* strains were included as controls. Colonies which showed a halo of clearing larger than the Δ*lasR* control strain were considered protease positive.

## Acyl homoserine lactone autoinducer bioreporter assays

Protocol as described in *Mould et al., 2020*. Briefly, 100 µL of $OD_{600 nm}$ normalized LB overnight cultures ($OD_{600 nm}$ = 0.01) of the AHL-synthesis deficient reporter strains DH161 (3OC12HSL-specific) or DH162 (3OC12HSL or C4HSL responsive) with AHL-responsive promoters to *lacZ* (*Whiteley and Greenberg, 2001*; *Whiteley et al., 1999*) were bead spread on LB plates containing 150 µg/mL 5-bromo-4-chloro-3-indolyl-β-D-galactopyranoside (XGAL, dissolved in DMSO). Inoculated plates were allowed to dry 10 min in a sterile hood. Once dry, 5 µL of either the test strains or control cultures

(PA14 wild type and Δ*lasR* strains) were spotted onto the inoculated reporter lawns. After the spots dried, plates were incubated at 37 °C for 16 hr then stored at 4 °C to allow for further color development, if necessary, based on wild-type colony activity. The blue halo that formed around the colony was interpreted as AHL activity. The levels of AHL produced are approximated by the size of the blue halo formed around the colony.

## Competition assays

Competition assays were performed by competing strains against an *att::lacZ* strain as previously reported (*Clay et al., 2020*). Overnight cultures of *att::lacZ* competitor and test strains were normalized to $OD_{600\ nm} = 1$ and mixed in the designated ratios with either a wild type control or Δ*lasR* strain. Aliquots of $10^{-6}$ dilutions of the initial mixed inoculums were immediately plated on LB plates containing 150 µg/mL XGAL by spreading an aliquot of 25–50 µL with sterilized glass beads. Roughly 100–200 colonies were counted to determine the initial ratios of PA14 *att:lacZ* to Δ*lasR* or the WT control strains by blue:white colony phenotype, respectively. To begin the competition experiment, a 250 µL aliquot of each undiluted mixed inoculum was sub-cultured into 5 mL fresh LB medium and incubated on a roller drum at 37 °C for 6 hr. After 6 hr, the cultures were collected, diluted by $10^{-6}$ in fresh liquid LB, and plated as previously stated for blue:white colony screening. The LB plates containing XGAL were incubated overnight at 37 °C prior to counting. Competitions were repeated on three separate days.

## Kinase mutant evolution screen

Using an ethanol/flame sterilized metal multiprong inoculation device (Dan-Kar), the kinase mutant library (*Wang et al., 2021*) was inoculated into a 96-well plate with 200 µL LB per well for 24 hr shaking at 37 °C. The 24 hr grown cultures were used to inoculate two 96-well plates with each kinase mutant (including PA14 WT control) in triplicate. These cultures were grown for 48 hr upon which 2 µL was transferred to new 96 well plates with fresh 200 µL LB liquid per well. Every 2 days, the wells containing the wild-type replicates were diluted by $10^{-6}$ in fresh LB and 25 µL was bead spread onto LB for phenotypic distinction based on sheen colony morphology. At Day 14, when all wildtype replicates contained at least 50% LasR⁻ phenotypes, all wells were diluted and plated as stated previously for determination of sheen colony morphology. A secondary screen in 5 mL LB (as described above in Experimental Evolution section) was initiated with those mutant strains which did not show any LasR⁻ phenotypes across all three replicates in the microtiter assay at Day 14. The Circos plot summarizing the screen data was generated using BioCircos (*Cui et al., 2016*) in R (version 4.0.2) and re-colored in Adobe Illustrator.

## Filtrate toxicity

Based on a protocol used previously (*Abisado et al., 2021*), strains were grown 16 hr in LB (5 mL) on a roller drum at 37 °C, centrifuged at 13 K RPM for 10 min in 2 mL aliquots, and the resulting supernatant was filter sterilized through a 0.22 µm pore filter. Per 5 mL filtrate, 250 µL of fresh LB was added. A 16 hr, grown LB culture (5 mL) of PA14 Δ*lasR* was normalized to an $OD_{600\ nm} = 1$ in LB, and 250 µL was used to inoculate 5 mL of the filtrate-LB mixture. The Δ*lasR* cultures were grown for 24 hr at 37 °C on the roller drum upon which colony counts were determined by bead spreading an appropriate dilution on LB plates. Data visualization and statistical analysis were performed in GraphPad Prism 9 (version 9.2.0).

## Fluoroacetamide sensitivity assay

Strains were inoculated (either by patching from plates or by spotting 5 µL of 16 h LB grown culture) onto plates containing 1.5% agar with M63 salts,10 mM lactamide, and 40 mM succinate with or without 2.5 mg/mL filter-sterilized fluoroacetamide (FAA) dissolved in water based on protocol by *Collier et al., 2001*. Relative growth was compared in the presence and absence of FAA. PA14 wild type and Δ*crc* were included as controls in every experiment wherein wild type displays robust growth on FAA in the presence of succinate and the Δ*crc* strain, little to none.

## Quantitative RT-PCR

The indicated strains were grown from single colonies in 5 mL LB cultures on a roller drum for 16 hr, normalized to an $OD_{600\ nm}$ of 1, and 250 µL of normalized culture was inoculated into 5 mL fresh LB for

a starting inoculum around $OD_{600\,nm}$ = 0.05. The cultures were then grown at 37 °C on a roller drum until $OD_{600\,nm}$ = 1 at which point a 1 mL aliquot of culture was pelleted by centrifugation for 10 min at 13 K RPM. Supernatant was removed, and the cell pellets were flash frozen in an ethanol dry ice bath. This was repeated on three separate days with one WT and one $\Delta lasR$ culture pair (n = 4) collected on each day or one $\Delta lasR$ and one $\Delta lasR\Delta cbrB$ culture pair (n = 3) each day. Pellets were stored at –80 °C until all sets of pellets were collected. RNA was extracted using the QIAGEN RNAeasy kit according to the manufacturer's protocol, and 7 µg RNA was twice DNAse treated with the Turbo DNA-free kit (Invitrogen). DNA contamination was checked by semi-quantitative PCR with gDNA standard for 35 cycles with *rpoD* qRT primers; if DNA contamination was greater than 0.004 ng / µL, the sample was DNAse treated again. cDNA was synthesized from 400 ng of DNase-treated RNA using the RevertAid H Minus first-strand cDNA synthesis kit (Thermo Scientific), according to the manufacturer's instructions for random hexamer primer (IDT) and a GC-rich template alongside an NRT control. Quantitative RT-PCR was performed on a CFX96 real-time system (Bio-Rad), using SsoFast Evergreen supermix (Bio-Rad) according to the following program: 95 °C for 30 s and 40 cycles of 95 °C for 5 s and 60 °C for 5 s followed by a melt curve with 65 °C for 3 s up to 95 °C in increments of 0.5 °C. Transcripts were normalized to the average *rpoD* and *rpsL* expression unless stated otherwise. *rpsL* and *crcZ* primers as designed in *Xia et al., 2020*. *rpoD* primers as designed in *Harty et al., 2019*. Data visualization and statistical analysis performed in GraphPad Prism 9 (version 9.2.0).

## Mono-carbon growth

Single carbon sources were supplemented into M63 base (*Neidhardt et al., 1974*) and filter sterilized. A 16 hr overnight LB culture grown at 37 °C on a roller drum was normalized to an $OD_{600\,nm}$ = 1 in 2 mL LB. For liquid growth curves, a 250 µL aliquot of the density adjusted culture was spiked into 5 mL fresh M63 medium with designated carbon source in triplicate, and growth was monitored using a Spectronic 20D+ (Spec20) hourly in 13 mm borosilicate tubes. Every point on the growth plots is the average of three replicates per day, repeated 3 days total. For colony biofilm growth, 5 µL of $OD_{600\,nm}$ = 1 normalized culture was inoculated onto 1.5% agar plate of M63 medium containing the designated carbon source in singlicate and grown for 16 hr at 37 °C. Colonies were cored using the back of a P1000 tip and disrupted by 5 min on Genie Disrupter in 1 mL LB. Disrupted colony biofilms were serially diluted. 5 µL of the serial dilutions were plated and a 50 µL aliquot of diluted colony resuspension ($10^{-6}$ or $10^{-7}$-fold, depending on condition/strain) was bead spread and counted for colony forming units. Colony biofilm growth was assessed on >5 independent days. Data visualization and statistical analysis performed in GraphPad Prism 9 (version 9.2.0).

## Metabolomics of bronchioloalveolar lavage fluid and artificial sputum medium

Human samples from people with and without cystic fibrosis were obtained with informed consent following institutional review board-approved protocols at Geisel School of Medicine at Dartmouth. The investigators were blinded to the conditions of the experiments during data collection and analysis. To obtain relative metabolite counts, bronchioloalveolar lavage (BAL) fluid samples were briefly centrifuged to exclude large debri then the supernatant was flash frozen in liquid nitrogen. Samples were processed by Metabolon via LC/MS for relative metabolite amounts. Raw values from Metabolon were normalized to protein concentrations by the BioRad Bradford protein concentration or raw area counts per day sample run and then the values were rescaled to set the median to one. Missing values were imputed with the minimum rescaled value for that biochemical. Quantitative amino acid concentrations were determined for aliquots of the same BAL samples (lyophilized) using the Biocrates AbsoluteIDQ p180 kit at the Duke Proteomics Core Facility. The lyophilized samples of BAL were homogenized in water and 50/50 water/methanol respectively to extract metabolites. 25 µL of the BAL extract were utilized for preparation of the samples on a Biocrates AbsoluteIDQ p180 plate. A Waters Xevo-TQ-S mass spectrometer was utilized to acquire targeted metabolite quantification on all samples and quality control specimens. Raw data (in µM) was exported independently for the FIA-MS/MS and UHPLC-MS/MS acquisition approaches used in this kit. The BAL sample data were corrected for the dilution factor since 25 µL was used versus 10 µL of the standards that were used to calculate the quantitative calibration curve. Principal component analysis of log normalized counts or concentrations were performed in R (version 4.0.2) (*R Development Core Team, 2021*) using the

prcomp() function and visualized with ggplot2 (*Wickham, 2016*) using ggfortify (*Tang et al., 2016*). *Supplementary file 3* of sample metadata was compiled with sjPlot (*Lüdecke, 2021*) in R.

## BIOLOG phenotyping assay

Two mL of LB overnight cultures grown at 37 °C on a roller drum were washed twice with M63 salts with no carbon source by repeated centrifugation (10 min, 13 K RPM) and resuspension into fresh medium. The washed cultures were normalized to an $OD_{600\,nm}$ = 0.05 in 25 mL of fresh M63 salts base and 100 µL was used to resuspend dehydrated carbon sources on the bottom of PM1 and PM2 BIOLOG phenotype plates by repeated pipetting. Cells and resuspended carbon were transferred to a sterile flat bottom, black-walled 96 well plate and incubated at 37 °C, static. Every hour $OD_{600\,nm}$ was monitored in a plate reader for 24 hr. Endpoint (24 hr) data is reported. Data visualization and statistical analysis performed in GraphPad Prism 9 (version 9.2.0).

## Acknowledgements

Research reported in this publication was supported by grants from the Cystic Fibrosis Foundation HOGAN19G0 (DAH), ASHARE20P0 (AA) and STANTO19R0 (DS), and the National Institutes of Health (NIH) through T32AI007519 (DLM), R01HL122372 (AA) and P20 GM130454-02 (DS). Additional core facility support came from the NIH NIGMS P20GM113132 (BioMT) and NIDDK P30-DK117469 (Dartmouth Cystic Fibrosis Research Center) and STANTO19R0 from the Cystic Fibrosis Foundation. Plasmid sequencing was carried out at Geisel School of Medicine Genomics Shared Resource, which was established by equipment grants from the NIH and NSF and is supported in part by a Cancer Center Core Grant (P30CA023108) from the NIH National Cancer Institute. We would like to acknowledge Amy Conaway for assistance in the kinase evolution screen, Dr. Georgia Doing for LasR⁻ colony enumeration in key experiments, and Dr. Nicholas Jacobs for constructive and thoughtful feedback on the written manuscript. The content is solely the responsibility of the authors and does not necessarily represent the official views of the NIH. The funders had no role in study design, data collection and analysis, decision to publish, or preparation of the manuscript.

## Additional information

### Funding

| Funder | Grant reference number | Author |
| --- | --- | --- |
| Cystic Fibrosis Foundation | HOGAN19G0 | Dallas L Mould<br>Deborah A Hogan |
| Cystic Fibrosis Foundation | ASHARE20P0 | Alix Ashare |
| Cystic Fibrosis Foundation | STANTO19R0 | Daniel Schultz |
| Cystic Fibrosis Foundation | T32AI007519 | Dallas L Mould |
| National Institutes of Health | R01HL122372 | Alix Ashare |
| National Institutes of Health | GM130454 | Mirjana Stevanovic<br>Daniel Schultz |
| National Institutes of Health | P20GM113132 | Dallas L Mould<br>Deborah A Hogan |
| National Institutes of Health | DK117469 | Dallas L Mould<br>Alix Ashare<br>Daniel Schultz<br>Deborah A Hogan |
| National Institutes of Health | P30CA023108 | Dallas L Mould<br>Alix Ashare<br>Daniel Schultz<br>Deborah A Hogan |

| Funder | Grant reference number | Author |
|---|---|---|

The funders had no role in study design, data collection and interpretation, or the decision to submit the work for publication.

## Author contributions
Dallas L Mould, Conceptualization, Data curation, Formal analysis, Investigation, Methodology, Supervision, Validation, Visualization, Writing - original draft, Writing – review and editing; Mirjana Stevanovic, Formal analysis, Methodology, Writing – review and editing; Alix Ashare, Funding acquisition, Resources, Writing – review and editing; Daniel Schultz, Conceptualization, Formal analysis, Funding acquisition, Methodology, Supervision, Visualization, Writing – review and editing; Deborah A Hogan, Conceptualization, Formal analysis, Funding acquisition, Project administration, Supervision, Validation, Writing – review and editing

## Author ORCIDs
Dallas L Mould ⓘ http://orcid.org/0000-0001-6939-1351
Deborah A Hogan ⓘ http://orcid.org/0000-0002-6366-2971

## Decision letter and Author response
Decision letter https://doi.org/10.7554/eLife.76555.sa1
Author response https://doi.org/10.7554/eLife.76555.sa2

---

# Additional files

## Supplementary files
• Supplementary file 1. Unfixed mutations observed in Pool-Seq data from evolved populations. Fixed mutations (i.e. those mutations present at 100% across all samples at all time points) were excluded from this list as potential background differences from reference strain. All unfixed mutations are listed with unique mutation ID that consists of gene name, mutation position on the genome and the nature of the mutation. ‡ symbol indicates multiple changes within the same codon. Gene identifiers, gene name and product name indicated in Columns 2–5 (i.e. B - E) for each unique mutation ID. Columns 6 through 14 (i.e. F - N) represent the difference in the fractions of sequenced reads for the listed variants from Day 4 to Day 6 (i.e. Day 6 fraction - Day 4 fraction) for each strain and replicate. Columns 14 - end (i.e. N - AE) represent the fraction of the sequenced reads with the listed variant. Columns are labeled by gene knockout_day_replicate_fraction or change.

• Supplementary file 2. Evolution screen for spontaneous LasR- strains in kinase deletion mutant backgrounds. Number of replicates (three possible) with no LasR- phenotypes observed at time of plating in LB microtiter evolution assay for each clean deletion mutant. The locus tag, gene length, start and stop genomic position, gene name indicated were used to generate Circos plot shown in *Figure 2—figure supplement 1*. Deletion mutants that displayed no LasR- phenotypes across all three replicates were followed up in a secondary 5 mL LB evolution assay.

• Supplementary file 3. Bronchioalveolar lavage (BAL) fluid donor characteristics from people with and without cystic fibrosis (CF) used for metabolomics analyses. Samples are grouped according to CF status with non-CF donors denoted 'HV' with age group, gender, CFTR genotype, and percent forced expiratory volume in 1 s at the time of encounter listed. nd, not determined.

• Supplementary file 4. Relative metabolite counts in bronchioalveolar lavage fluid from people with cystic fibrosis (CF) and non-CF comparators by LC/MS. Values listed for each biochemical are normalized and imputed as described in the methods section. The percent forced expiratory volume in 1 s is noted for samples from people with CF as part of the column title. For Non-CF samples, column labels are listed numerically with 'Non-CF' indicator. Counts are shown per sample, as the average across CF (n = 10) and non-CF (n = 10) samples and as ratio of the averages between CF and non-CF as well. Information on (sub)pathway, biochemical identifiers, and methodology shown per biochemical.

• Supplementary file 5. Amino acid and biogenic amine concentrations in bronchioalveolar lavage (BAL) fluid from people with cystic fibrosis (CF) and non-CF comparators. Values listed are micromolar concentrations for specified metabolites quantified in the BAL samples from people with and without CF (non-CF) using the Biocrates AbsoluteIDQ p180 Kit (n = 10 and 2, respectively).

The percent forced expiratory volume in 1 s is noted for samples from people with CF as part of the column title. In addition to the concentrations listed per sample for each amino acid, the average is shown for samples from people with and without CF as well as a ratio of the averages.

• Supplementary file 6. Strains and plasmids used in this study. List of strains and plasmids used in this study with internal lab strain identifier, short description of strain/use including gene name and gene number, and source listed.

• Supplementary file 7. Supplemental methods for the choice of parameters for modeling wild type and LasR- frequencies in passaged cultures. Description of the terms for lag, growth, and death for both wild type strain PA14 and derived *lasR* mutants, and estimates of the number of *lasR* mutants in the initial population.

• Transparent reporting form

• Source code 1. Matlab script for deterministic mathematical model of LasR+ (wild type) strain PA14 and lasR mutant growth following density and percentage of each strain over the course of evolution regime as described in methods. Model parameters include growth rate, carrying capacity/concentration of saturated culture, killing rate (lysis-growth), difference in lag time (hours), and killing time at the end (hours) for each strain. Code output used to generate Figure 1A.

### Data availability

All sequencing data is available on the Sequence Read Archive with accession number PRJNA786588 upon publication. All data generated or analyzed and all code used during this study are included in the manuscript or associated files.

The following dataset was generated:

| Author(s) | Year | Dataset title | Dataset URL | Database and Identifier |
|---|---|---|---|---|
| Mould DL | 2021 | Pool Seq of Experimentally Evolved *P. aeruginosa* PA14 populations in LB | https://www.ncbi.nlm.nih.gov/sra/?term=PRJNA786588 | NCBI Sequence Read Archive, PRJNA786588 |

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

## Appendix 1

### Appendix 1—key resources table

| Reagent type (species) or resource | Designation | Source or reference | Identifiers | Additional information |
|---|---|---|---|---|
| Gene (*Pseudomonas aeruginosa*) | PA14 | NCBI | Accession: GCF_000014625.1 | Reference file |
| Strain, strain background (*Saccharomyces cerevisiae*) | *Saccharomyces cerevisiae* | PMID:16820502 | | Cloning yeast |
| Strain, strain background (*Escherichia coli*) | DH5a | Invitrogen | | Electrocompetent cells |
| Strain, strain background (*Escherichia coli*) | S17 $\lambda$ pir | PMID:8226632 | | Electrocompetent cells made in lab |
| Strain, strain background (*P. aeruginosa*) | PA14 WT | PMID:7604262 | | Hogan Laboratory reference strain |
| Strain, strain background (*P. aeruginosa*) | DH2417; NC-AMT0101-1-2 | PMID:16687478 | | Chronic CF lung infection isolate with functional LasR allele |
| Strain, strain background (*P. aeruginosa*) | PA14 WT | PMID:33771779 | | Laub Lab; strain background of kinase clean deletion mutants |
| Genetic reagent (*P. aeruginosa*) | PA14 Δ*lasR* | PMID:15554963 | | |
| Genetic reagent (*P. aeruginosa*) | PAO-MW1qsc102 | PMID:10570171; PMID:11544214 | | AHL-sensing bioreporter; PAO1 *lasIrhlI* mutant with Tn5-B22, which contains promoterless *lacZ* located within PA1896 (hypothetical protein) at chromosomal location of 2,067,716. Responsive to 3OC$_{12}$-HSL but not C$_4$-HSL. |
| Genetic reagent (*P. aeruginosa*) | PAO-MW1qsc131 | PMID:10570171; PMID:11544214 | | AHL-sensing bioreporter; PAO1 *lasIrhlI* mutant with Tn5-B22, which contains promoterless *lacZ*, under *phzC* promoter control. Responsive to either 3OC$_{12}$-HSL or C$_4$-HSL but requires both for full activation. |
| Genetic reagent (*P. aeruginosa*) | PA14 WT *att::lacZ* | PMID:31980538 | | PA14 WT with constitutive expression of *lacZ* for competition assays |
| Genetic reagent (*P. aeruginosa*) | PA14 Δ*cheA* | PMID:33771779 | | |
| Genetic reagent (*P. aeruginosa*) | PA14 Δ*chpA* | PMID:33771779 | | |
| Genetic reagent (*P. aeruginosa*) | PA14 Δ*creC* | PMID:33771779 | | |
| Genetic reagent (*P. aeruginosa*) | PA14 Δ*uhpB* | PMID:33771779 | | |
| Genetic reagent (*P. aeruginosa*) | PA14 Δ*bfiS* | PMID:33771779 | | |
| Genetic reagent (*P. aeruginosa*) | PA14 Δ*bphP* | PMID:33771779 | | |
| Genetic reagent (*P. aeruginosa*) | PA14 Δ*PA14_10770* | PMID:33771779 | | |
| Genetic reagent (*P. aeruginosa*) | PA14 Δ*PA14_11630* | PMID:33771779 | | |

*Appendix 1 Continued on next page*

*Appendix 1 Continued*

| Reagent type (species) or resource | Designation | Source or reference | Identifiers | Additional information |
|---|---|---|---|---|
| Genetic reagent (*P. aeruginosa*) | PA14 Δ*rocS1* | PMID:33771779 | | |
| Genetic reagent (*P. aeruginosa*) | PA14 Δ*narX* | PMID:33771779 | | |
| Genetic reagent (*P. aeruginosa*) | PA14 Δ*wspE* | PMID:33771779 | | |
| Genetic reagent (*P. aeruginosa*) | PA14 Δ*PA14_19340* | PMID:33771779 | | |
| Genetic reagent (*P. aeruginosa*) | PA14 Δ*mxtR* | PMID:33771779 | | |
| Genetic reagent (*P. aeruginosa*) | PA14 Δ*cpxS* | PMID:33771779 | | |
| Genetic reagent (*P. aeruginosa*) | PA14 Δ*gtrS* | PMID:33771779 | | |
| Genetic reagent (*P. aeruginosa*) | PA14 Δ*PA14_24340* | PMID:33771779 | | |
| Genetic reagent (*P. aeruginosa*) | PA14 Δ*rocS2* | PMID:33771779 | | |
| Genetic reagent (*P. aeruginosa*) | PA14 Δ*PA14_26810* | PMID:33771779 | | |
| Genetic reagent (*P. aeruginosa*) | PA14 Δ*sagS* | PMID:33771779 | | |
| Genetic reagent (*P. aeruginosa*) | PA14 Δ*copS* | PMID:33771779 | | |
| Genetic reagent (*P. aeruginosa*) | PA14 Δ*pfeS* | PMID:33771779 | | |
| Genetic reagent (*P. aeruginosa*) | PA14 Δ*bqsS* | PMID:33771779 | | |
| Genetic reagent (*P. aeruginosa*) | PA14 Δ*PA14_30700* | PMID:33771779 | | |
| Genetic reagent (*P. aeruginosa*) | PA14 Δ*PA14_30840* | PMID:33771779 | | |
| Genetic reagent (*P. aeruginosa*) | PA14 Δ*czcS* | PMID:33771779 | | |
| Genetic reagent (*P. aeruginosa*) | PA14 Δ*PA14_32570* | PMID:33771779 | | |
| Genetic reagent (*P. aeruginosa*) | PA14 Δ*PA14_36420* | PMID:33771779 | | |
| Genetic reagent (*P. aeruginosa*) | PA14 Δ*ercS* | PMID:33771779 | | |
| Genetic reagent (*P. aeruginosa*) | PA14 Δ*exaD* | PMID:33771779 | | |
| Genetic reagent (*P. aeruginosa*) | PA14 Δ*ercS'* | PMID:33771779 | | |
| Genetic reagent (*P. aeruginosa*) | PA14 Δ*parS* | PMID:33771779 | | |
| Genetic reagent (*P. aeruginosa*) | PA14 Δ*kdpD* | PMID:33771779 | | |
| Genetic reagent (*P. aeruginosa*) | PA14 Δ*PA14_43670* | PMID:33771779 | | |

*Appendix 1 Continued on next page*

*Appendix 1 Continued*

| Reagent type (species) or resource | Designation | Source or reference | Identifiers | Additional information |
|---|---|---|---|---|
| Genetic reagent (*P. aeruginosa*) | PA14 ΔPA14_45590 | PMID:33771779 | | |
| Genetic reagent (*P. aeruginosa*) | PA14 ΔPA14_45870 | PMID:33771779 | | |
| Genetic reagent (*P. aeruginosa*) | PA14 ΔPA14_46370 | PMID:33771779 | | |
| Genetic reagent (*P. aeruginosa*) | PA14 ΔPA14_46980 | PMID:33771779 | | |
| Genetic reagent (*P. aeruginosa*) | PA14 ΔPA14_48160 | PMID:33771779 | | |
| Genetic reagent (*P. aeruginosa*) | PA14 Δ*phoQ* | PMID:33771779 | | |
| Genetic reagent (*P. aeruginosa*) | PA14 ΔPA14_49420 | PMID:33771779 | | |
| Genetic reagent (*P. aeruginosa*) | PA14 Δ*fleS* | PMID:33771779 | | |
| Genetic reagent (*P. aeruginosa*) | PA14 Δ*pirS* | PMID:33771779 | | |
| Genetic reagent (*P. aeruginosa*) | PA14 Δ*gacS* | PMID:33771779 | | |
| Genetic reagent (*P. aeruginosa*) | PA14 Δ*tctE* | PMID:33771779 | | |
| Genetic reagent (*P. aeruginosa*) | PA14 Δ*pprA* | PMID:33771779 | | |
| Genetic reagent (*P. aeruginosa*) | PA14 Δ*colS* | PMID:33771779 | | |
| Genetic reagent (*P. aeruginosa*) | PA14 ΔPA14_57170 | PMID:33771779 | | |
| Genetic reagent (*P. aeruginosa*) | PA14 Δ*roxS* | PMID:33771779 | | |
| Genetic reagent (*P. aeruginosa*) | PA14 Δ*rcsC* | PMID:33771779 | | |
| Genetic reagent (*P. aeruginosa*) | PA14 Δ*pvrS* | PMID:33771779 | | |
| Genetic reagent (*P. aeruginosa*) | PA14 Δ*pilS* | PMID:33771779 | | |
| Genetic reagent (*P. aeruginosa*) | PA14 Δ*cbrA* | PMID:33771779 | | |
| Genetic reagent (*P. aeruginosa*) | PA14 Δ*pmrB* | PMID:33771779 | | |
| Genetic reagent (*P. aeruginosa*) | PA14 Δ*retS* | PMID:33771779 | | |
| Genetic reagent (*P. aeruginosa*) | PA14 ΔPA14_64580 | PMID:33771779 | | |
| Genetic reagent (*P. aeruginosa*) | PA14 Δ*aruS* | PMID:33771779 | | |
| Genetic reagent (*P. aeruginosa*) | PA14 Δ*ntrB* | PMID:33771779 | | |
| Genetic reagent (*P. aeruginosa*) | PA14 ΔPA14_68230 | PMID:33771779 | | |

*Appendix 1 Continued*

| Reagent type (species) or resource | Designation | Source or reference | Identifiers | Additional information |
|---|---|---|---|---|
| Genetic reagent (*P. aeruginosa*) | PA14 Δ*amgS* | PMID:33771779 | | |
| Genetic reagent (*P. aeruginosa*) | PA14 Δ*algZ* | PMID:33771779 | | |
| Genetic reagent (*P. aeruginosa*) | PA14 Δ*phoR* | PMID:33771779 | | |
| Genetic reagent (*P. aeruginosa*) | PA14 Δ*kinB* | PMID:33771779 | | |
| Genetic reagent (*P. aeruginosa*) | PA14 ΔPA14_72740 | PMID:33771779 | | |
| Genetic reagent (*P. aeruginosa*) | PA14 Δ*rhlR* | PMID:30936375 | | |
| Genetic reagent (*P. aeruginosa*) | PA14 Δ*anr* | PMID:31527114 | | |
| Genetic reagent (*P. aeruginosa*) | PA14 Δ*cbrB* | This paper | | PA14 WT with in-frame deletion of *cbrB* (PA14_62540) |
| Genetic reagent (*P. aeruginosa*) | PA14 *cbrB*::Tn*M* | PMID:22911607; PMID:16477005 | | *cbrB* MAR2xT7 transposon insertion mutant |
| Genetic reagent (*P. aeruginosa*) | DH2417Δ*cbrB*; NC-AMT0101-1-2Δ*cbrB* | This paper | | CF clinical isolate NC-AMT0101-1-2 (DH2417) with in-frame deletion of *cbrB* (PA14_62540) |
| Genetic reagent (*P. aeruginosa*) | PA14 Δ*crc* | This paper | | PA14 WT with in-frame deletion of *crc* (PA14_70390) |
| Genetic reagent (*P. aeruginosa*) | PA14 Δ*crc* + *crc* | This paper | | PA14 Δ*crc* with complementation of *crc* (PA14_70390) at the native locus |
| Genetic reagent (*P. aeruginosa*) | PA14 Δ*lasR* + *lasR* | PMID:31980538 | | |
| Genetic reagent (*P. aeruginosa*) | PA14 Δ*cbrB* + pMQ70 *cbrB* (plasmid) | This paper | | PA14 Δ*cbrB* expressing arabinose inducible pMQ70 *cbrB* expression vector |
| Genetic reagent (*P. aeruginosa*) | PA14 Δ*lasR*Δ*cbrB* | This paper | | PA14 Δ*lasR* with in-frame deletion of *cbrB* (PA14_62540) |
| Genetic reagent (*P. aeruginosa*) | PA14 Δ*lasR*Δ*cbrB* + *cbrB* | This paper | | PA14 Δ*lasR*Δ*cbrB* with complementation of *cbrB* (PA14_62540) at the native locus |
| Genetic reagent (*P. aeruginosa*) | PA14 Δ*lasR*Δ*cbrB*Δ*crc* | This paper | | PA14 Δ*lasR*Δ*cbrB* with in-frame deletion of *crc* (PA14_70390) |
| Genetic reagent (*P. aeruginosa*) | PA14 Δ*lasR*Δ*crc* | This paper | | PA14 Δ*lasR* with in-frame deletion of *crc* (PA14_70390) |
| Genetic reagent (*P. aeruginosa*) | PA14 Δ*lasR*Δ*crc* + *crc* | This paper | | PA14 Δ*lasR*Δ*crc* with complementation of *crc* (PA14_70390) at the native locus |
| Genetic reagent (*P. aeruginosa*) | PA14 + pMQ72 EV | This paper | | PA14 WT expressing pMQ72 empty vector |
| Genetic reagent (*P. aeruginosa*) | PA14 + pMQ70 *cbrB* | This paper | | PA14 WT expressing arabinose inducible pMQ70 *cbrB* expression vector |
| Genetic reagent (*P. aeruginosa*) | PA14 + pMQ72 *crcZ* | This paper | | PA14 WT expressing arabinose inducible pMQ72 *crcZ* expression vector |
| Recombinant DNA reagent | pMQ30 EV | PMID:16820502 | | |
| Recombinant DNA reagent | pMQ72 EV | PMID:16820502 | | |
| Recombinant DNA reagent | pMQ70 EV | PMID:16820502 | | |

*Appendix 1 Continued on next page*

*Appendix 1 Continued*

| Reagent type (species) or resource | Designation | Source or reference | Identifiers | Additional information |
|---|---|---|---|---|
| Recombinant DNA reagent | pMQ30_*cbrB*_KO | This paper | | pMQ30 plasmid for knocking out *cbrb* |
| Recombinant DNA reagent | pMQ30_*crc*_KO | This paper | | pMQ30 plasmid for knocking out *crc* |
| Recombinant DNA reagent | pMQ30_*cbrB*_KON | This paper | | pMQ30 plasmid for complementing *cbrb* at the native locus |
| Recombinant DNA reagent | pMQ30_*crc*_KON | This paper | | pMQ30 plasmid for complementing *cbrb* at the native locus |
| Recombinant DNA reagent | pMQ72_*crcZ* | This paper | | pMQ72 plasmid backbone with arabinose inducible *crcZ* expression |
| Recombinant DNA reagent | pMQ70_*cbrB* | This paper | | pMQ70 plasmid backbone with arabinose inducible *cbrB* expression |
| Sequence-based reagent | pMQ72 crcZ OE 1 F | This paper | PCR Primers; construct design | gtttctccatacccgttttttgggctagcGCACAACAACAATAACAAGCAACGACGAAG |
| Sequence-based reagent | pMQ72 crcZ OE 2 R | This paper | PCR Primers; construct design | ctagaggatccccgggtaccgagctcgaattcgaaatggtgtaaggcgaaggaaaaacgg |
| Sequence-based reagent | pMQ70 *cbrB* OE 1 F | This paper | PCR Primers; construct design | ctctctactgtttctccatacccgttttttgggctagcgAGACGAGCgaattcACGTCGAGAGAGCtgaatacatggcac |
| Sequence-based reagent | pMQ70 *cbrB* OE 2 R | This paper | PCR Primers; construct design | ttgcatgcctgcaggtcgactctagaggatccccgggtacGTAACAGGTTGCAGGGTaccGTtacgagtcggccgaggcccc |
| Sequence-based reagent | pMQ30 *cbrB* KO 1 F | This paper | PCR Primers; construct design | taacaatttcacacaggaaacagctatgaccatgattacgaattcAGGAAGTGCTGATGTGGAACC |
| Sequence-based reagent | pMQ30 *cbrB* KO 2 R | This paper | PCR Primers; construct design | GTAACAGGTTGCAGGGTGTTTATTCAGCTCTCTCGACGTGCT |
| Sequence-based reagent | pMQ30 *cbrB* KO 3 F | This paper | PCR Primers; construct design | CACGTCGAGAGAGCTGAATAAACACCCTGCAACCTGTTACC |
| Sequence-based reagent | pMQ30 *cbrB* KO 4 R | This paper | PCR Primers; construct design | aggtcgactctagaggatccccgggtaccgagctcgaattcCAGGGAGTGCTGGTTGTTACCGATGACTtc |
| Sequence-based reagent | pMQ30 *crc* KO 1 F | This paper | PCR Primers; construct design | taacaatttcacacaggaaacagctatgaccatgattacgaattcTGGAATACAGGCGCAGCAac |
| Sequence-based reagent | pMQ30 *crc* KO 2 R | This paper | PCR Primers; construct design | TAGAAAAGCCGGCGCATGCGCTGGCTTTTTCGTGTCTGACGGGGCAAATGGCCCCCAAAATCACGTGCG |
| Sequence-based reagent | pMQ30 *crc* KO 4 R | This paper | PCR Primers; construct design | ctgcaggtcgactctagaggatccccgggtaccgagctcgaattcttggctgaccgccgagtacggcatgc |
| Sequence-based reagent | pMQ30 *crc* KO 3 F | This paper | PCR Primers; construct design | TTTGAGCTCGGGTATCATACACGCACGTGATTTTGGGGGCCATTTGCCCCGTCAGACACGAAAAAGCCAG |
| Sequence-based reagent | *cbrB* check F | This paper | PCR primer, KO check | GCGTCTGCTCCCTGGCCAAG |
| Sequence-based reagent | *cbrB* check R | This paper | PCR primer, KO check | GTGGCGCTGGTGGCGACATC |
| Sequence-based reagent | *crc* check F | This paper | PCR primer, KO check | GCTCGATGGCGAAACGAATG |
| Sequence-based reagent | *crc* check R | This paper | PCR primer, KO check | GCGCTGGTGTTGACCATCATC |
| Sequence-based reagent | crcZ RT 1 F | PMID:31911486 | PCR Primers | GCACAACAACAATAACAAGCAACG |
| Sequence-based reagent | crcZ RT 2 R | PMID:31911486 | PCR Primers | AGTTTTATTCTTCTTCCGACTGGCT |

*Appendix 1 Continued on next page*

*Appendix 1 Continued*

| Reagent type (species) or resource | Designation | Source or reference | Identifiers | Additional information |
|---|---|---|---|---|
| Sequence-based reagent | rpsL RT1F | PMID:31911486 | PCR Primers | GTAAGGTATGCCGTGTACG |
| Sequence-based reagent | rpsL RT 2 R | PMID:31911486 | PCR Primers | CACTACGCTGTGCTCTTG |
| Sequence-based reagent | rpoD RT 1 F | PMID:30936375 | PCR Primers | CGCCGAGATCAAGGAAATCA |
| Sequence-based reagent | rpoD RT 2 R | PMID:30936375 | PCR Primers | TACTTCTTGGCGATGGAAATCA |
| Commercial assay, kit | NEBuilder HiFi DNA Assembly | Biolabs | Cat. #: E2621L | Gibson cloning |
| Commercial assay, kit | Master Pure Yeast DNA purification kit | Lucigen | Cat. No.: MPY80200 | |
| Commercial assay, kit | RNAeasy Mini kit | QIAGEN | Cat. No.: 74,104 | |
| Commercial assay, kit | Turbo DNA-free kit | Thermo Fisher Scientific | AM1907 | |
| Commercial assay, kit | RevertAid H Minus First Strand cDNA synthesis | Thermo Scientific | Cat. No.: EP0451 | cDNA synthesis with IDT random hexamer |
| Commercial assay, kit | SsoFast EvaGreen Supermix | BIO-RAD | Cat.#: 1725201 | |
| Commercial assay, kit | Zymoprep Yeast Plasmid Miniprep II | Zymo Research | Cat. No.: D2004 | Yeast cloning |
| Commercial assay, kit | Biocrates AbsoluteIDQ p180 kit | biocrates | | Amino acid. quantification |
| Chemical compound, drug | Brain Heart Infusion | BD | SKU:211,059 | BBL Brain Heart Infusion |
| Chemical compound, drug | Agar | BD | SKU:214,510 | Difco Agar, granulated (2 Kg pail) |
| Chemical compound, drug | XGAL; 5-bromo-4-chloro-3-indolyl-β-D-galactopyranoside | Research Products International | B71800-5.0 | Stock dissolved in DMSO |
| Chemical compound, drug | Fluoroacetamide | Aldrich | 128341–5 G | dissolved in water, filter sterilized |
| Chemical compound, drug | Lactamide | Tokyo Chemical Industry | Product Number: L0001 | |
| Chemical compound, drug | Mannitol | Sigma | M8429-500G | D-Mannitol |
| Chemical compound, drug | Succinate | Sigma | S9512-500G | Succinic acid, to pH7 with NaOH |
| Chemical compound, drug | Phenylalanine | Sigma-Aldrich | P2126-100G | L-Phenylalanine |
| Chemical compound, drug | Arginine | Sigma | A5131-100G | L-arginine monohydrochloride |
| Chemical compound, drug | Lactate | Fisher Scientific | S326-500 | Sodium lactate syrup, 60% w/w |
| Chemical compound, drug | Glucose | VWR Chemicals | BDH9230-2.5KG | Dextrose, anhydrous |
| Chemical compound, drug | Citrate | FisherScientific | A104-500 | Citric acid, monohydrate |

*Appendix 1 Continued*

| Reagent type (species) or resource | Designation | Source or reference | Identifiers | Additional information |
|---|---|---|---|---|
| Chemical compound, drug | Gentamicin | Research Products International | G38000-10.0 | Gentamicin Sulfate |
| Chemical compound, drug | Nalidixic acid | Research Products International | N42000-25.0 | |
| Chemical compound, drug | Carbinicillin | Goldbio | C-103–25 | Carbenicillin (Disodium) |
| Chemical compound, drug | Sucrose | Fisher BioReagents | BP220-212; 2.5 kg | D-Sucrose |
| Chemical compound, drug | HEPES | SIGMA | H3375-100G | buffer |
| Chemical compound, drug | Tryptone | Fisher Bioreagents | BP1421-500 | LB |
| Chemical compound, drug | NaCl; sodium chloride | Fisher Chemical | S271-3 | LB |
| Chemical compound, drug | Yeast extract | Fisher Bioreagents | BP1422-500 | LB |
| Chemical compound, drug | Ammonium sulfate | Fisher Chemical | A702-500 | M63 |
| Chemical compound, drug | Potassium phosphate monobasic | Fisher Chemical | P382-500 | M63 |
| Chemical compound, drug | Potassium phosphate dibasic, anhydrous | Fisher Chemical | P288-500 | M63 |
| Chemical compound, drug | Magnesium Sulfate | Fisher Scientific | M63-500 | M63 |
| Chemical compound, drug | Milk | BD | 232,100 | Difco Skim Milk |
| Chemical compound, drug | Yeast Nitrogen Base without amino acids | Research Products International | Y20040-500.0 | Yeast cloning |
| Chemical compound, drug | Yeast Synthetic Drop-out Medium Supplements without uracil | Sigma | Y1501-20G | Yeast cloning |
| Chemical compound, drug | Peptone | Fisher bioreagents | BP1420-500 | YPD |
| Chemical compound, drug | Sodium phosphate dibasic anhydrous | Fisher Chemicals | S374-500 | ASM |
| Chemical compound, drug | Sodium phosphate monobasic | Fisher Chemicals | S369-500 | ASM |
| Chemical compound, drug | Potassium Nitrate | Fisher Chemicals | M-12636 | ASM |
| Chemical compound, drug | Potassium Sulfate | Fisher Chemicals | P304-500 | ASM |
| Chemical compound, drug | L-(+)-Lactic acid | Sigma | L1750-10G | ASM; 1 M stock; pH to 7 with NaOH |
| Chemical compound, drug | Calcium chloride dihydrate | Sigma | C7902-500G | ASM |
| Chemical compound, drug | Magnesium Chloride Hexahydrate | Fisher chemical | M33-500 | ASM |

*Appendix 1 Continued on next page*

*Appendix 1 Continued*

| Reagent type (species) or resource | Designation | Source or reference | Identifiers | Additional information |
|---|---|---|---|---|
| Chemical compound, drug | FeSO$_4$*7H$_2$O | Sigma-Aldrich | F-8048 | ASM; Ferrous sulfate heptahydrate; Filter sterilized |
| Chemical compound, drug | N-acetylglucosamine | Fisher Scientific | AAA1304718 | ASM |
| Chemical compound, drug | DPPC | Sigma | P0763-250MG | ASM; 1,2-dipalmitoyl-sn-glycero-3-phosphocholine; dissolved in chloroform |
| Chemical compound, drug | Tryptophan | Sigma-Aldrich | T0254-25G | ASM, L-Tryptophan |
| Chemical compound, drug | Mucin | Sigma | M2378-100G | ASM; Mucin from porcine stomach, Type II |
| Peptide, recombinant protein | Phusion | New England BioLabs | M0530L | High-Fidelity DNA polymerase |
| Software, algorithm | MATLAB | MathWorks | | |
| Software, algorithm | breseq | PMID:24838886 | | Version 0.35.4 |
| Software, algorithm | bcl2fastq | Illumina | RRID:SCR_015058 | v2.20.0422 |
| Software, algorithm | GraphPad Prism 9 | GraphPad | | Version 9.2.0 |

