## [Editor Report]

This study aimed to identify the genetic foundation favoring selection of lasR mutants in laboratory and clinical isolates from persons with CF. They selected these mutants using a predictable and quantitative framework of evolution experiments and then identified their genetic underpinnings by a suppressor screen. The role of cbrAB as a key intermediate is important and ties together several reports of nutrient-dependent advantages of lasR, including those that may explain their adaptation to conditions found in the CF airway.

---

## [Decision Letter]

**Decision letter after peer review:**

Thank you for submitting your article "Metabolic basis for the evolution of a common pathogenic *Pseudomonas aeruginosa* variant" for consideration by *eLife*. Your article has been reviewed by 3 peer reviewers, one of whom is a member of our Board of Reviewing Editors, and the evaluation has been overseen by Gisela Storz as the Senior Editor. The following individual involved in review of your submission has agreed to reveal their identity: Karin Kram (Reviewer #3).

Essential revisions:

1) Please clarify the open questions of the complexity of the CF nutritional environment selecting for lasR mutants, which are also selected in multiple, simpler environments.

2) It's unclear whether the QS (dys-)function of lasR- is necessary to explain these results; mutants appear to be selected independent of a social interaction with producers. Please address.

3) Figure 1E: The authors state that the model predicted the extent to which LasR- outcompeted the WT, but the model appears to have predicted slightly higher percentages for the LasR- strain than observed in the experiment. Is this difference significant? Should this finding be interpreted as evidence that other phenotypes of LasR- mutants beyond metabolic adaptations contribute to some extent to their fitness?

4) Line 213/Figure 2A: Although the selection of LasR- mutants was delayed in the CF mutant (DH2417), LasR- phenotypes still consistently occurred and rose to high frequencies. This is a major point that suggests that CbrB may play a role but is not required for LasR- mutant fitness in certain genetic backgrounds. The wording in line 213 and conclusions throughout ought to carefully reflect this point.

5) Line 404: It is curious that crc mutations occur in a ∆cbrB background but are not frequently observed in vivo or in a WT background. While it is true that the effect of a ∆crc mutant is not identical the effect of a lasR mutation, the ∆crc mutant still appears to produce many similar phenotypes but to a different magnitude. While this pathway is required for LasR- mutant fitness, is it possible that phenotypes unrelated to this pathway also contribute to LasR- mutant fitness, enabling their selection?

6) It is not clear whether or why ∆anr or ∆rhlR strains are used to compare rates of LasR- mutations.

*Reviewer #2 (Recommendations for the authors):*

This is an exceptionally thorough and well-written manuscript. Comments for the authors are indicated below.

Figure 1E: The authors state that the model predicted the extent to which LasR- outcompeted the WT, but the model appears to have predicted slightly higher percentages for the LasR- strain than observed in the experiment. Is this difference significant? Should this finding be interpreted as evidence that other phenotypes of LasR- mutants beyond metabolic adaptations contribute to some extent to their fitness?

Line 213/Figure 2A: Although the selection of LasR- mutants was delayed in the CF mutant (DH2417), LasR- phenotypes still consistently occurred and rose to high frequencies. This is a major point that suggests that CbrB may play a role but is not required for LasR- mutant fitness in certain genetic backgrounds. The wording in line 213 and conclusions throughout ought to carefully reflect this point.

Figure 4: Do the authors have experimental data to model growth parameters of lasR mutants in their ASM medium? This would be interesting to compare with the LB data in Figure 1.

Line 404: It is curious that crc mutations occur in a ∆cbrB background but are not frequently observed in vivo or in a WT background. While it is true that the effect of a ∆crc mutant is not identical the effect of a lasR mutation, the ∆crc mutant still appears to produce many similar phenotypes but to a different magnitude. While this pathway is required for LasR- mutant fitness, is it possible that phenotypes unrelated to this pathway also contribute to LasR- mutant fitness, enabling their selection?

Line 548: Does read trimming refer to trimming of Illumina adapters, to other quality-control related trimming procedures, or both? Please briefly indicate the software and parameters used for this step.

Line 552: Was variant calling analysis performed using the -p option to call polymorphisms? If so, please add this detail to the example command.

Supplemental Figure 6B: Lines should not connect the data points in this figure as the x-axis is categorical.

*Reviewer #3 (Recommendations for the authors):*

I might mention quorum sensing/the function of LasR somewhere in the abstract, especially since that is the opening paragraph in the introduction.

Figure 1A: For the model, why OD600 instead of CFU/ml or cells?

Figure 1B: y-axis should just be % LasR phenotype (since it shows both + and -).

Figure 1E is not particularly clear nor do I think this experiment adds much, since you already show that LasR- mutants can outcompete even when starting as a very low percentage of the population.

Line 244: citation twice.

Line 289: maybe you could add something to the end of this sentence like, "retains the control of Crc-Hfq mediated regulation, allowing for Crc expression, even when crcZ is repressed." The way this sentence is written currently is a bit hard to decipher.

Line 294: Do you know nothing else increases crcZ expression?

Figure 4C: It is really difficult to determine which metabolites are important or significant (the way it is right now it looks like all are labeled as significant, but that can't be true). Maybe just show the ones that are significantly different here (or a subset), and the rest in a supplemental figure? You don't explain the pvalue in the legend, or the colors.

I am not sure the experiment in 4D is particularly additive, since the LasR- phenotype does not evolve faster or more dramatically in LB. It indicates that these can still evolve in ASM, but there is nothing specific about ASM. Also, it should be clear in the text that this is the same passaging scheme as was used previously.

Why do you think no gain of function mutations, or those that lead to increased expression, are found for cbrA/cbrB? Why there may be no loss-of-function crc mutations is briefly addressed.

---

## [Author Response]

Essential revisions:1) Please clarify the open questions of the complexity of the CF nutritional environment selecting for lasR mutants, which are also selected in multiple, simpler environments.

We agree that this would be a useful addition to the text. We have added the following sentence to the Introduction.

“While it is clear that there are many ways in which LasR loss-of-function mutants differ from their LasR+ progenitors, a common trait that promotes the rise of LasR– strains in diverse environments, even in rich and minimal laboratory media (Heurlier et al., 2005; Scribner, Stephens, Huong, Richardson, and Cooper, 2022) (O'Brien, Luján, Paterson, Cant, and Buckling, 2017) (Lujan, Moyano, Segura, Argarana, and Smania, 2007; Qi, Toll-Riera, Heilbron, Preston, and MacLean, 2016; Robitaille, Groleau, and Déziel, 2020; Sandoz et al., 2007; Wong, Rodrigue, and Kassen, 2012; Yan et al., 2018), has not been established.”

We also added text to the Discussion that highlights how the complex CF lung environment may provide multiple positive and negative selective pressures.

“Unlike *lasR* mutations, *crc* mutations are not commonly observed in clinical isolates (Winstanley, O'Brien, and Brockhurst, 2016) and *crc* mutants have been shown to be under negative selection in Tn-Seq experiments (Lorenz et al., 2019).”

2) It's unclear whether the QS (dys-)function of lasR- is necessary to explain these results; mutants appear to be selected independent of a social interaction with producers. Please address.

Overall, our data suggest that LasR- selection is independent of social interaction and driven by growth differences under the conditions assessed in this paper. The concordance between the modeling described in Figure 1A, which does not include any terms for positive or negative interactions, and the actual data shown in Figure 1B is one of the major pieces of data in support of this conclusion. We have revised our description of the motivation for the model in the revised manuscript.

“We built a mathematical model of strain competition based exclusively on experimentally-determined mono-culture growth parameters to predict the relative changes in wild type and LasR– cell numbers when grown on a common pool of growth substrates to determine how differences in growth kinetics alone would impact the rise of LasR– lineages (Figure 1A).”

We have also revised the description of Figure 1E to indicate that it does not show density dependent fitness for LasR strains under our conditions, which suggests that “cheating” does not play a driving role for the improved growth of LasR- *P. aeruginosa* in co-culture.

“When the ∆*lasR* competitor was cultured with the tagged wild type for 6 h, the percentage of ∆*lasR* mutant cells in the total population increased regardless of the initial percentage of ∆*lasR* (1% to 85%) at the time of inoculation (Figure 1E). The model successfully predicted that ∆*lasR* would outcompete the wild type over this range which is consistent with differential growth kinetics playing a major role (Figure 1E-dotted line). No ∆*lasR* advantage would be observed when it is at high initial percentages if its advantage was solely due to exploitation of common goods, as is observed when WT and ∆*lasR* are co-cultured on a substrate that requires WT protease production (Sandoz et al., 2007). There were differences between the best fit lines for the actual and predicted data that could be due to a variety of factors including measurement error or biological interactions between WT and ∆*lasR* strains (e.g. policing Castañeda-Tamez et al., 2018; M. Wang et al., 2015).”

We have also added the following text to the manuscript discussion:

“The overlap between the model-predicted and observed percentages of LasR^–^ strains over the course of the evolution regime suggests that the described social advantage resulting from QS dysfunction for LasR^–^ strains (i.e. social cheating) is not necessary to explain the timing and kinetics of the initial rise of LasR^–^ strains under our conditions. However, social interactions that benefit LasR- strains may be evident where the model estimates percentages that fall below or on the lower range of that experimentally observed, such as Day 4 or 6. A more detailed discussion of when social interactions are required is found below.”

3) Figure 1E: The authors state that the model predicted the extent to which LasR- outcompeted the WT, but the model appears to have predicted slightly higher percentages for the LasR- strain than observed in the experiment. Is this difference significant? Should this finding be interpreted as evidence that other phenotypes of LasR- mutants beyond metabolic adaptations contribute to some extent to their fitness?

In the revised the description of Figure 1E, we discuss the fact that the model predicts higher percentages of LasR- strains than observed. We speculate that this could be due to decreased fitness of LasR- strains in the presence of LasR+ strains. This may relate to “policing” mechanisms by which wild type strains kill LasR- strains through a variety of secreted goods as a way of controlling the levels which LasR- strains reach. We have added text to this effect to the discussion.

4) Line 213/Figure 2A: Although the selection of LasR- mutants was delayed in the CF mutant (DH2417), LasR- phenotypes still consistently occurred and rose to high frequencies. This is a major point that suggests that CbrB may play a role but is not required for LasR- mutant fitness in certain genetic backgrounds. The wording in line 213 and conclusions throughout ought to carefully reflect this point.

We agree that in LB the loss of *cbrB* delayed and reduced the percentage of LasR- strains in the CF isolate, but that LasR- colonies still arose (Figure 2B), and that we did not accurately note this in the referenced sentence. We have corrected this.

In addition, we added the following text to the discussion:

“It is interesting to note that there were differences in the relative dependence on CbrB for the selection for LasR^–^ between strains PA14 and DH2417, and in ASM the dependence on *cbrB* for the selection of LasR^–^ strains increased in both strains (Figure 4D) suggesting that different environments may alter the importance of different LasR and CbrB controlled targets important for fitness that have yet to be elucidated.

5) Line 404: It is curious that crc mutations occur in a ∆cbrB background but are not frequently observed in vivo or in a WT background. While it is true that the effect of a ∆crc mutant is not identical the effect of a lasR mutation, the ∆crc mutant still appears to produce many similar phenotypes but to a different magnitude. While this pathway is required for LasR- mutant fitness, is it possible that phenotypes unrelated to this pathway also contribute to LasR- mutant fitness, enabling their selection?

We have modified text to address this interesting point as follows:

“Because CbrAB activity can still be suppressed by succinate in LasR^–^ cells (Figure 2E), LasR^–^ variants were not strictly “de-repressed”, and this is consistent with the fact that ∆*lasR* and ∆*crc* growth patterns were not identical. Unlike *lasR* mutations, *crc* mutations are not commonly observed in clinical isolates (Winstanley, O'Brien, and Brockhurst, 2016) and *crc* mutants have been shown to be under negative selection in Tn-Seq experiments (Lorenz et al., 2019).”

6) It is not clear whether or why ∆anr or ∆rhlR strains are used to compare rates of LasR- mutations.

We revised the text to clarify the motivation for these experiments as follows:

“LasR^–^ strains rose with a similar frequency as in the wild type progenitor when evolution experiments were initiated with strains lacking the regulator Anr, important for LasR^–^ microoxic fitness, or the regulator RhlR, important for *lasR* mutant policing (Chen, Déziel, Groleau, Schaefer, and Greenberg, 2019; Clay et al., 2020) suggesting that these regulators were not major contributors to fitness under these conditions (Figure 2 —figure supplement 2B).”

Reviewer #2 (Recommendations for the authors):This is an exceptionally thorough and well-written manuscript. Comments for the authors are indicated below.Lines 49 and 80: Inconsistencies in citation format are present in a few instances.

Thank you. We think that this refers to the fact that the *eLife* Endnote output style only lists one author when the author list is long.

Figure 1E: The authors state that the model predicted the extent to which LasR- outcompeted the WT, but the model appears to have predicted slightly higher percentages for the LasR- strain than observed in the experiment. Is this difference significant? Should this finding be interpreted as evidence that other phenotypes of LasR- mutants beyond metabolic adaptations contribute to some extent to their fitness?

This comment was included in essential comments and is addressed above.

Line 213/Figure 2A: Although the selection of LasR- mutants was delayed in the CF mutant (DH2417), LasR- phenotypes still consistently occurred and rose to high frequencies. This is a major point that suggests that CbrB may play a role but is not required for LasR- mutant fitness in certain genetic backgrounds. The wording in line 213 and conclusions throughout ought to carefully reflect this point.

This comment was included in essential comments and is addressed above.

Figure 2C: It is not clear what the arrows are pointing at.

To improve clarity, we have removed the arrows and added detail about the larger colony size in text and figure legend.

Figure 4: Do the authors have experimental data to model growth parameters of lasR mutants in their ASM medium? This would be interesting to compare with the LB data in Figure 1.

We agree that this would be interesting. Because ASM is not optically clear, we cannot measure cellular numbers using the same parameters/tools used for LB (OD600). The code included in this manuscript for the mathematical model (Source Code File 1) can be used by the community and the parameters altered for growth in other conditions.

Line 404: It is curious that crc mutations occur in a ∆cbrB background but are not frequently observed in vivo or in a WT background. While it is true that the effect of a ∆crc mutant is not identical the effect of a lasR mutation, the ∆crc mutant still appears to produce many similar phenotypes but to a different magnitude. While this pathway is required for LasR- mutant fitness, is it possible that phenotypes unrelated to this pathway also contribute to LasR- mutant fitness, enabling their selection?

This comment was included in essential comments and is addressed above.

Line 548: Does read trimming refer to trimming of Illumina adapters, to other quality-control related trimming procedures, or both? Please briefly indicate the software and parameters used for this step.

Read trimming was performed with bcl2fastq (v2.20.0422) to remove Illumina adaptor sequences during the demultiplexing process. This information has been added to the methods.

Line 552: Was variant calling analysis performed using the -p option to call polymorphisms? If so, please add this detail to the example command.

Variant calling analysis was performed using the -p option to call polymorphisms. We have added this detail to the example command.

Supplemental Figure 6B: Lines should not connect the data points in this figure as the x-axis is categorical.

Agreed. We have removed the line.

Reviewer #3 (Recommendations for the authors):I might mention quorum sensing/the function of LasR somewhere in the abstract, especially since that is the opening paragraph in the introduction.

Thank you. We have added text on the function of LasR to the abstract.

Figure 1A: For the model, why OD600 instead of CFU/ml or cells?

Given that the model was built on data obtained as OD600, as a proxy for cell number, we maintained this as an output for the model. We note that OD600 correlates with cell number shown in panel B.

Figure 1B: y-axis should just be % LasR phenotype (since it shows both + and -).

Thank you. We have corrected the y-axis in Figure 1B and Figure 2F as well.

Figure 1B legend should explain the differences in the symbols (not just the text).

We have added this detail to the legend.

Figure 1E is not particularly clear nor do I think this experiment adds much, since you already show that LasR- mutants can outcompete even when starting as a very low percentage of the population.

We revised the text describing this figure. We feel that it is useful to compare to other studies that assess social interactions, which we discuss in more detail in this manuscript (see Essential Comment 3).

Line 289: maybe you could add something to the end of this sentence like, "retains the control of Crc-Hfq mediated regulation, allowing for Crc expression, even when crcZ is repressed." The way this sentence is written currently is a bit hard to decipher.

Thank you for this suggestion. We have altered the text and figure to highlight the point that *lasR* mutants respond to succinate while *crc* mutants do not.

Line 294: Do you know nothing else increases crcZ expression?

Because we can’t rule out players other than CbrAB in the control of crcZ, we have modified this sentence to say:

“CbrAB activity induces the expression of *crcZ*, which sequesters Crc. We found that the ∆*lasR* mutant had ~ two-fold higher *crcZ* levels compared to wild type, suggesting higher activity of the CbrAB two component system in LasR^–^ strains (Figure 3A).”

Figure 4C: It is really difficult to determine which metabolites are important or significant (the way it is right now it looks like all are labeled as significant, but that can't be true). Maybe just show the ones that are significantly different here (or a subset), and the rest in a supplemental figure? You don't explain the pvalue in the legend, or the colors.

In this figure we are not comparing individual carbon sources, but rather the relative density of WT and ∆*lasR* across those carbon sources present on BIOLOG PM-1 and PM-2 plates that were enriched in the CF lung. We have added additional details to the text, legend and figure for clarification. Thank you.

I am not sure the experiment in 4D is particularly additive, since the LasR- phenotype does not evolve faster or more dramatically in LB. It indicates that these can still evolve in ASM, but there is nothing specific about ASM. Also, it should be clear in the text that this is the same passaging scheme as was used previously.

We retained this figure and now discuss that while the kinetics for the increase in LasR- are similar in LB and ASM, it is interesting that the dependence on *cbrB* for LasR- lineage selection (particularly for strain 2417) is greater in ASM. We have added this observation to the discussion to highlight how environment may affect fitness as discussed above (see Essential Comment 4).

We have added text to indicate that the evolution scheme is similar.

Why do you think no gain of function mutations, or those that lead to increased expression, are found for cbrA/cbrB? Why there may be no loss-of-function crc mutations is briefly addressed.

There is one mutation found in an upstream region of CbrB, though the impact of this mutation on expression is currently unknown. It would be interesting if it were a gain of function in terms of expression.

We have expanded our discussion of the negative fitness of *crc* single mutants in the revised discussion, and believe that they only frequently rise to detectable levels in *cbrB* mutant populations.